# Regulation of meiotic telomere dynamics through membrane fluidity promoted by AdipoR2-ELOVL2

Jingjing Zhang[1], Mario Ruiz[1], Per-Olof Bergh [2], Marcus Henricsson [2], Nena Stojanović[1], Ranjan Devkota[1], Marius Henn[3], Mohammad Bohlooly-Y[4], Abrahan Hernández-Hernández[5,7], Manfred Alsheimer [3], Jan Borén [2], Marc Pilon [1] ✉ & Hiroki Shibuya [1,6] ✉

The cellular membrane in male meiotic germ cells contains a unique class of phospholipids and sphingolipids that is required for male reproduction. Here, we show that a conserved membrane fluidity sensor, AdipoR2, regulates the meiosis-specific lipidome in mouse testes by promoting the synthesis of sphingolipids containing very-long-chain polyunsaturated fatty acids (VLC-PUFAs). AdipoR2 upregulates the expression of a fatty acid elongase, ELOVL2, both transcriptionally and post-transcriptionally, to synthesize VLC-PUFA. The depletion of VLC-PUFAs and subsequent accumulation of palmitic acid in *AdipoR2* knockout testes stiffens the cellular membrane and causes the invagination of the nuclear envelope. This condition impairs the nuclear peripheral distribution of meiotic telomeres, leading to errors in homologous synapsis and recombination. Further, the stiffened membrane impairs the formation of intercellular bridges and the germ cell syncytium, which disrupts the orderly arrangement of cell types within the seminiferous tubules. According to our findings we propose a framework in which the highly-fluid membrane microenvironment shaped by AdipoR2-ELOVL2 underpins meiosis-specific chromosome dynamics in testes.

During mitosis, the cytoplasm of the dividing cell forms a temporal bridging structure called the midbody[1]. Midbody formation in mitotic cells is regulated not only by a series of midbody-localizing proteins[2], but also by the local rearrangement of the lipid membrane that enriches classical sphingolipids as well as their glycosylated variants (glycosphingolipids) (as summarized in Supplementary Fig. 1)[3]. The inhibition of glycosphingolipid biosynthesis leads to defective cytokinesis[4], suggesting that the rearrangement of the lipid membrane is crucial for faithful cytokinesis.

While the mitotic midbody is a temporal structure, dividing germ cells do not complete cytokinesis and instead form stable intercellular bridge structures resulting in the formation of germline syncytia[5]. In mammalian germ cells, the intercellular bridge is composed of canonical midbody proteins in addition to a germ cell-specific component

[1]Department of Chemistry and Molecular Biology, University of Gothenburg, 41390 Gothenburg, Sweden. [2]Department of Molecular and Clinical Medicine/ Wallenberg Laboratory, Institute of Medicine, University of Gothenburg, 41467 Gothenburg, Sweden. [3]Department of Cell and Developmental Biology, Biocenter, University of Würzburg, 97074 Würzburg, Germany. [4]Discovery Sciences, BioPharmaceutical R&D, AstraZeneca, Gothenburg, Sweden. [5]Department of Cell and Molecular Biology, Karolinska Institutet, Stockholm, Sweden. [6]Laboratory for Gametogenesis, RIKEN Center for Biosystems Dynamics Research (BDR), Kobe, Japan. [7]Present address: National Genomics Infrastructure, Science for Life Laboratory, Department of Cell and Molecular Biology, Karolinska Institute, Stockholm, Sweden. ✉e-mail: marc.pilon@cmb.gu.se; hiroki.shibuya@riken.jp

called Testis Expressed 14 (TEX14). TEX14 stabilizes the temporal midbody structure and forms the stable intercellular bridge by inhibiting the recruitment of proteins essential for cell abscission (Supplementary Fig. 1a)[6]. The deletion of *Tex14* and the subsequent disruption of the intercellular bridge results in the death of prophase I spermatocytes and male infertility in mice[7]. In contrast, female mice with a deletion of *Tex14* exhibit a disruption of intercellular bridge structures while maintaining fertility, suggesting the sexually dimorphic requirement of the germline syncytium[8].

Accumulating evidence suggests that the formation of the intercellular bridge in mammalian germ cells requires specialized forms of sphingolipids that contain very-long-chain polyunsaturated fatty acids (VLC-PUFAs) (Supplementary Fig. 1b)[9–13]. VLC-PUFAs are not present in ordinary dietary sources and thus need to be synthesized in situ. VLC-PUFAs are uniquely synthesized in the brain, retina, and testis and have specialized roles in these tissues such as brain physiology, photoreceptor activities, and male reproduction, respectively[14]. In the testis, the synthesis of VLC-PUFA-containing sphingolipids requires the germ cell-specific ceramide synthase CerS3 as well as the fatty acid (FA) elongase ELOVL2[9,10]. CerS3 deficiency results in the disorganization of the intercellular bridge and thus the death of prophase I spermatocytes, similar to *Tex14* knockout (KO) mice. Notably, TEX14 still localizes to the disorganized intercellular bridge in *CerS3* KO spermatocytes, suggesting that lipid rearrangement is essential for intercellular bridge stability in addition to the TEX14-dependent regulation of midbody proteins. However, the detailed molecular regulations underlying the VLC-PUFA synthesis in the testis and, more crucially, which aspects of meiosis require VLC-PUFAs are poorly understood.

During meiotic prophase I, homologous chromosomes are paired and recombined[15]. To achieve the homology search, chromosomes attach to the nuclear envelope (NE) via their telomeres and move along the NE[16,17]. The misregulation of telomere-driven chromosome movements leads to failures in pairing and recombination and to human infertility, including azoospermia and primary ovarian insufficiency[18–21]. Even though the responsible molecular regulations of meiotic telomere-binding proteins have been clarified, how the membrane is regulated for this purpose remains unexplained. Telomeres are mechanically fused with the lipid bilayer and subsequently move along the membrane[22], thus there must be a need for the rearrangement of lipid species to fluidize the membrane.

Here, we report that the highly-fluid lipid environment ensured by the synthesis of VLC-PUFA in male germ cells is required for the nuclear peripheral distribution of meiotic telomeres and thus for faithful homologous recombination and synapsis. The synthesis of VLC-PUFA in testes depends on AdipoR2 (Adiponectin receptor 2), an evolutionarily conserved regulator of membrane homeostasis[23–26]. AdipoR2 ensures the robust expression of ELOVL2 in the testis. Our findings establish an essential role for AdipoR2-ELOVL2 in shaping the highly-fluid membrane microenvironment, thus allowing the chromosome dynamics needed for proper meiosis progression in male germlines.

## Results

### AdipoR2 is essential for male reproduction
AdipoR2, as well as its paralog AdipoR1, were initially identified as the receptors for Adiponectin, a secretory protein produced by adipocytes[23]; however, recent studies have demonstrated their prime role as evolutionarily conserved regulators of membrane homeostasis[24–26]. To analyze their tissue-specific roles, we used reverse transcription PCR (RT-PCR) to assess mRNA expressions in murine tissue samples. *AdipoR1* and *AdipoR2* were widely expressed in somatic tissues, while *AdipoR2* alone demonstrated predominant expression in the testis (Fig. 1a).

To investigate *AdipoR2*'s roles in the testis, we analyzed spermatogenesis defects in *AdipoR2* KO testes (Fig. 1b). The KO testes were significantly smaller compared to those of their WT littermates (Fig. 1c). Histological staining showed that there were only a small number of post-meiotic round spermatids in the KO seminiferous tubules and that elongated spermatids were completely absent (Supplementary Fig. 2). Of note, the round spermatids in the KO testis were abnormally large and multinucleated, and these cells were undergoing apoptosis (Fig. 1d, e), which is similar to the defects reported in lipid metabolism-defective mutants such as *CerS3* KO and *Elovl2* KO testes[9,10]. Consistent with the histological analysis, transmission electron microscopy (TEM) analysis found that multiple round spermatid nuclei lacking the discernable acrosome structure resided in a large cytoplasmic area enclosed by a plasma membrane (Fig. 1f). Consistent with the testicular defects, the epididymides from *AdipoR2* KO males were devoid of sperm (Fig. 1g), which led to the complete loss of male fertility (Fig. 1h).

### *AdipoR2* deficiency causes homologous synapsis and recombination failures
The TUNEL assay showed that a large number of germ cells underwent apoptosis in mid-pachytene spermatocytes at epithelial cycle stage V–VIII (Fig. 2a). To investigate meiotic defects further, we analyzed the progression of meiotic prophase I by staining with SYCP1 (a homologous synapsis marker) and SYCP3 (an axial element marker) (Supplementary Fig. 3a). Quantification revealed a significant increase in leptotene (unsynapsed) and zygotene (partially synapsed) populations in the *AdipoR2* KO testis (Fig. 2b). Closer inspection of *AdipoR2* KO pachytene spermatocytes showed the presence of two major synapsis defects—unsynapsis (represented by a few chromosomes remaining unsynapsed along the whole length of their chromosome axes even when the other chromosomes have completely synapsed) or incomplete synapsis (represented by abnormal Y-shaped branching structures) (Fig. 2c).

During meiosis, unsynapsed chromatin activates a response termed meiotic silencing of unsynapsed chromatin (MSUC) and leads to the accumulation of DNA-damage markers such as the phosphorylated serine 139 of histone H2AX (γH2AX)[27,28]. In WT pachytene cells, the γH2AX signal was restricted to sex chromosomes and formed one flare signal on the condensed sex chromosomes (Fig. 2d). In contrast, there was a significant increase in the number of γH2AX flares in the *AdipoR2* KO pachytene spermatocytes. The chromatin near the branching axis was stained with γH2AX, suggesting that MSUC was activated at the incompletely synapsed regions (Fig. 2d, magnified pictures). The RAD51 recombinase was not detected at the γH2AX flares in the *AdipoR2* KO pachytene spermatocytes, suggesting that it is not unrepaired DNA double-strand breaks (DSBs) but MSUC that is responsible for γH2AX flare formation (Supplementary Fig. 3b). Further, thickened and condensed SCYP3-stained chromosome axes seen on the sex chromosomes (the so-called "sex body") in WT spermatocytes were barely distinguishable in the *AdipoR2* KO pachytene spermatocytes (Fig. 2d, rightmost panel), suggesting that the activation of MSUC led to imperfect silencing and to the decondensation of sex chromosomes.

Meiotic homologous recombination is facilitated by homologous synapsis, thus failures in homologous synapsis frequently coincide with defects in homologous recombination[29,30]. The counting of RPA2 foci, a single-strand DNA-binding protein that decorates the DSB sites, and the type I crossover marker MutL homolog 1 (MLH1) in *AdipoR2* KO spermatocytes, revealed defects in DSB repair and crossover formation (Fig. 2e, f). Taken together, *AdipoR2* depletion significantly impairs homologous chromosome synapsis and recombination, which causes aberrant activation of MSUC on autosomes leading to the massive, if not complete, elimination of male germ cells at the mid-pachytene stage.

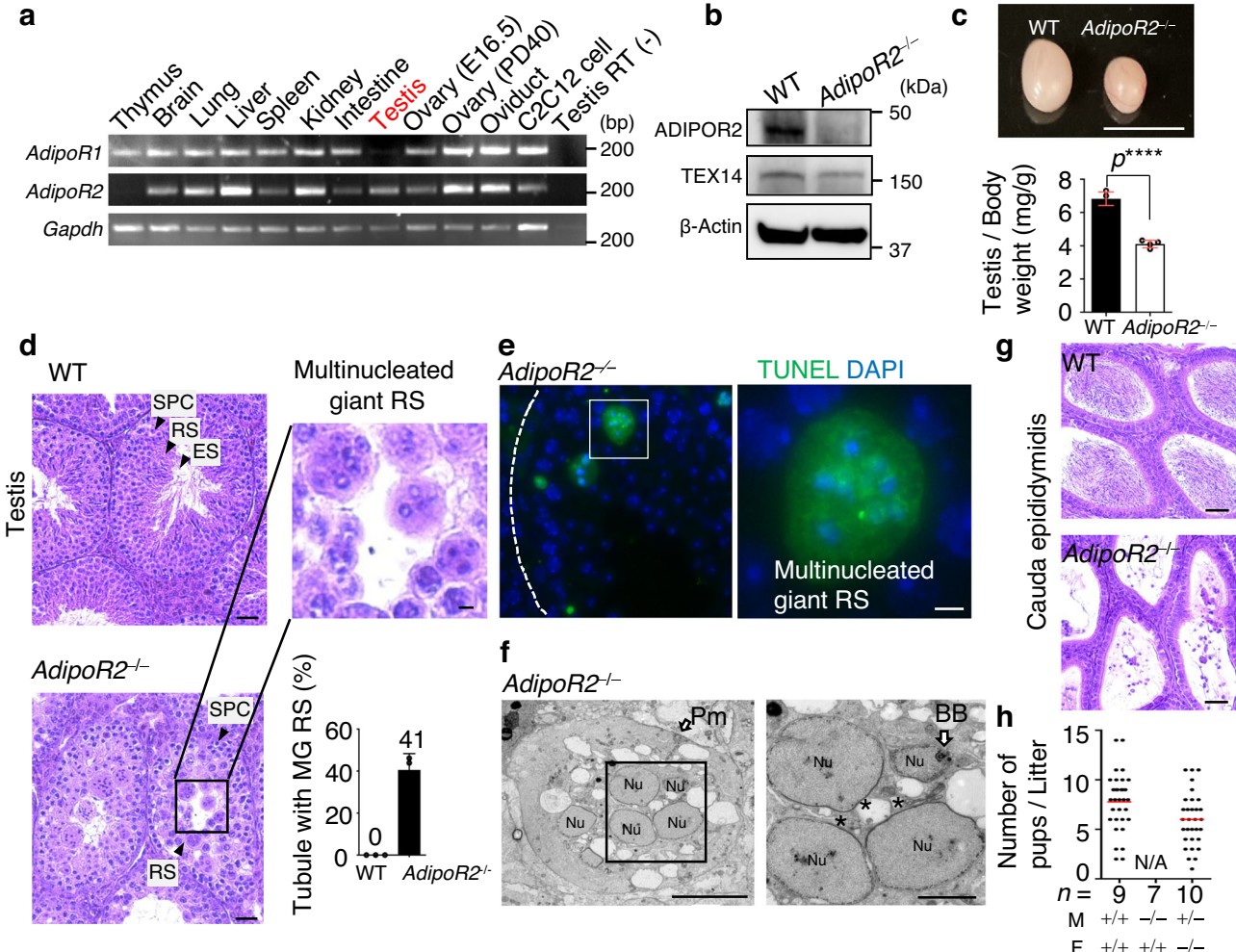

**Fig. 1 | AdipoR2 is indispensable for male reproduction. a** Tissue-specific expression of *AdipoR1, AdipoR2*, and *Gapdh* (loading control) shown by RT-PCR. C2C12 is a myoblast cell line. E; embryonic day. PD; postnatal day. The testis sample without the RT reaction was used as the negative control. Experiments were repeated in a total of 2 mice. **b** Immunoblots of mouse testis extracts from WT and *AdipoR2⁻/⁻* male mice. Experiments were repeated in a total of 3 mice. **c** Testes from 8-week-old WT and *AdipoR2⁻/⁻* male mice. Scale bar: 1 cm. The graph shows the testis/body weight ratio. Mean values with SD are shown. Four mice were analyzed for each genotype. ****p < 0.0001 by two-tailed *t*-tests. **d** Testis sections from 8-week-old WT and *AdipoR2⁻/⁻* males stained with hematoxylin and eosin. The arrowheads indicate a spermatocyte (SPC), round spermatid (RS), and elongated spermatid (ES). Scale bars: 50 μm (10 μm in the magnified panel). The graph shows the frequency of seminiferous tubules containing the multinucleated giant RS. Mean values with SD are shown. Three mice were analyzed for each genotype, and more than 900 seminiferous tubules were analyzed for each mouse. **e** Testis sections from 8-week-old WT and *AdipoR2⁻/⁻* male mice stained with TUNEL and DAPI. Scale bars: 10 μm. Experiments were repeated in a total of 3 mice. **f** TEM images of multinucleated giant round spermatids in *AdipoR2⁻/⁻* testes. Pm: plasma membrane. Nu: nucleus. BB: basal body. The asterisks indicate the thickened NE at the apical side, where no regular acrosome assembly was found. Scale bars: 5 μm (2 μm in the magnified panel). Experiments were repeated in a total of 3 mice. **g** Epididymis sections from 8-week-old WT and *AdipoR2⁻/⁻* males were stained with hematoxylin and eosin. No mature sperm were present in the *AdipoR2⁻/⁻* epididymis. Scale bar: 50 μm. Experiments were repeated in a total of 3 mice. **h** The average number of pups per litter. PD60 male (M) and female (F) WT (+/+), *AdipoR2⁻/⁻*, and *AdipoR2⁺/⁻* mice were paired for more than 90 days of continuous breeding. *n* indicates the number of mating pairs examined. Source data are provided as a Source Data file.

## AdipoR2 is essential for the synthesis of testis-specific VLC−PUFA-containing sphingolipids

To gain insight into the mechanism underlying the meiotic errors, we performed a whole-testis lipidomics analysis to determine whether the FA composition is changed in *AdipoR2* KO testis. To minimize the heterogeneity in the cell population between WT and *AdipoR2* KO testes, we used juvenile testis samples at postnatal day 28. We determined the composition of two dominant phospholipid classes, phosphatidylcholine (PC) and phosphatidylethanolamine (PE), as well as of the sphingolipid ceramide (Fig. 3). Consistent with previous studies[9,10], the WT testes samples were enriched in PC, PE, and ceramides containing VLC-PUFAs, which is a unique feature of the testis lipidome (Fig. 3a). However, the *AdipoR2* KO testes showed significant accumulation of FAs containing shorter carbon chains such as PC 32:0, PC 34:1, PE 34:1, and ceramide 16:0 (Cer 16:0)

and a significant reduction of VLC-PUFAs such as PC 36:4, PC 38:5, PC 38:6, PE 36:4, ceramide 30:6, and ceramide 30:5 (Fig. 3a). The most significantly affected lipid species was Cer 16:0, a sphingosine anchored with a palmitic acid, which was 22 mol % more over-represented in *AdipoR2* KO testes compared to WT (Fig. 3b). Similar trends were detected even when the lipids were quantified in terms of nmol or pmol/mg tissue (Supplementary Fig. 4). The heat map visualization showed a clear gradient where the saturated or monounsaturated FAs with shorter carbon chains were over-represented and, conversely, the VLC-PUFAs were mostly depleted in the *AdipoR2* KO samples (Fig. 3c). The analysis of another sphingolipid species, sphingomyelin, showed the same trends (Supplementary Fig. 5). These results suggest that AdipoR2 is essential for the synthesis of testis-specific VLC-PUFA−containing phospholipids and sphingolipids.

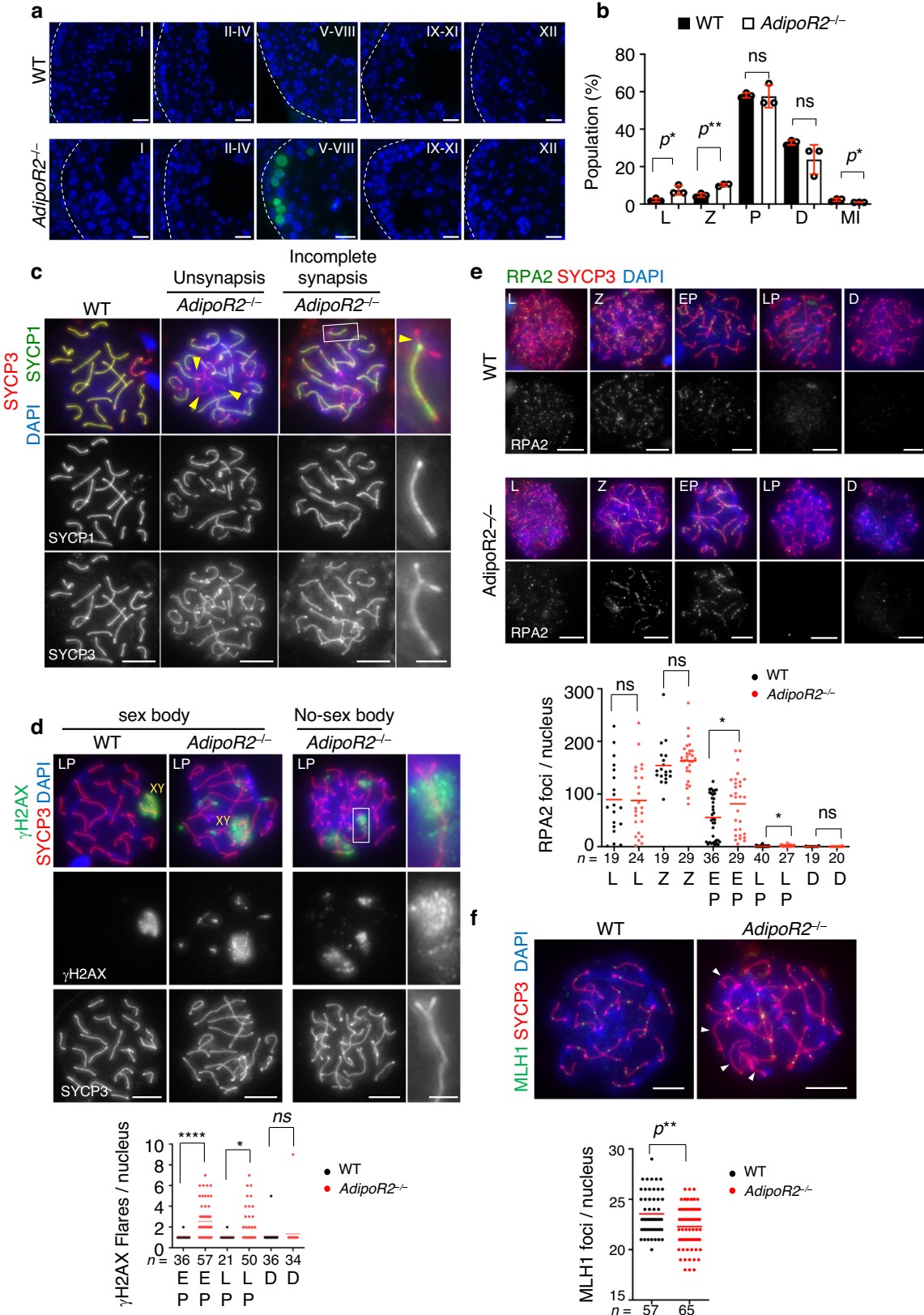

### AdipoR2 regulates ELOVL2 at both the transcriptional and post-transcriptional levels

VLC-PUFA−containing ceramide is synthesized through multiple steps, including FA desaturation by desaturases (such as FADS1, FADS2, and SCD1), FA elongation by elongases (such as ELOVL2, ELOVL4, and ELOVL5), and the transfer of the acyl chain from acyl-CoA to a sphingoid base by ceramide synthase (such as testis-specific CerS3) (Fig. 4a). To address the mechanism underlying VLC-PUFA−containing ceramide synthesis by AdipoR2, we examined the mRNA expression of key enzymes by quantitative PCR (qPCR). Among the candidates tested, the mRNA expression of *Elovl2* and *CerS3* was significantly decreased in *AdipoR2* KO testes compared to WT (Fig. 4b and

**Fig. 2 | AdipoR2 ensures the synapsis and recombination of homologous chromosomes. a** Testis sections from 8-week-old WT and *AdipoR2*[−/−] male mice stained with TUNEL and DAPI. Scale bars: 10 µm. **b** The populations of each meiotic prophase substage in testis cell suspensions from 8-week-old WT and *AdipoR2*[−/−] male mice. Representative images are shown in Fig. S3A. SYCP3-positive spermatocytes (1454 cells for WT and 1,475 cells for *AdipoR2* KO pooled from three mice for each genotype) were classified into the following substages: L, leptotene (no SYCP1); Z, zygotene (partially assembled SYCP1); P, pachytene (fully assembled SYCP1); D, diplotene (disassembled SYCP1); and MI, metaphase I (SYCP3 at centromeres). The mean values of three independent experiments with SD are shown. Two-tailed *t*-tests. ns: not significant. L $p = 0.024$, Z $p = 0.0016$, MI $p = 0.045$. **c** Immunostaining of pachytene spermatocytes from WT and *AdipoR2*[−/−] male mice. Arrowheads: asynapsis. The magnified image highlights the incompletely synapsed chromosome with a Y-shaped branch between aligned homologs (arrowhead). Scale bars: 5 µm (1 µm in the magnified panel). Experiments were repeated in a total of 3 mice. **d** Immunostaining of pachytene spermatocytes from WT and *AdipoR2*[−/−]

male mice. Sex chromosomes (XY) distinguished by condensed SYCP3 signals are indicated. The magnified picture highlights the incompletely synapsed chromosome with a Y-shaped branch that accompanied a γH2AX focus. The graph shows the number of γH2AX flares with mean values. *n* shows the analyzed spermatocyte number pooled from three mice for each genotype. EP, early pachytene; LP, late pachytene; D, diplotene. Two-tailed *t*-tests. ns: not significant. *$p = 0.019$, ****$p < 0.0001$. Scale bars: 5 µm (1 µm in the magnified panel). **e** Immunostaining of spermatocytes from WT and *AdipoR2*[−/−] male mice. Scale bars: 5 µm. The graph shows the mean number of RPA2 foci associated with the chromosome axes. *n* shows the analyzed spermatocyte number pooled from three mice for each genotype. Two-tailed *t*-tests. ns: not significant. EP $p = 0.034$, LP $p = 0.016$. **f** Immunostaining of late pachytene spermatocytes from WT and *AdipoR2*[−/−] male mice. Chromosomes without MLH1 foci are highlighted by arrowheads. Scale bars: 5 µm. The graph shows the number of MLH1 foci per late pachytene spermatocyte with mean values. *n* shows the analyzed cell number pooled from three mice. Two-tailed *t*-tests. **$p = 0.0011$. Source data are provided as a Source Data file.

*CerS1,2,4,5,6* data in Supplementary Fig. 6a). Consistent with this, Western blotting analysis showed a moderate reduction of CerS3 protein levels and an almost complete abrogation of ELOVL2 protein expression in *AdipoR2* KO testes (Fig. 4c, d). These results suggest that AdipoR2 regulates ELOVL2 expression both at the transcriptional and post-transcriptional levels. The lipidomics analysis showed that the total amount of ceramide, normalized to the total amount of PC, was not significantly changed in the *AdipoR2* KO lipidome compared to WT (Supplementary Fig. 6b). Further, we found comparable localization of CerS3 protein in endoplasmic reticulum membranes in mid-pachytene spermatocytes as reported in previous studies (Supplementary Fig. 6c)[9]. These results suggest that the moderate reduction of CerS3 protein has only a minor effect on the bulk ceramide level in the *AdipoR2* KO lipidome. Instead, the almost complete loss of ELOVL2 protein expression in *AdipoR2* KO testes readily explains the depletion of VLC-PUFA in the *AdipoR2* KO lipidome.

## Membrane rigidification impairs the nuclear peripheral distribution of meiotic telomeres

To investigate the consequences of the depletion of VLC-PUFA in the *AdipoR2* KO testis lipidome, we quantitatively measured the fluidity of the cellular membranes. Isolated live male germ cells were cultured in vitro and stained with Laurdan dye to measure membrane packing (often a proxy measurement for fluidity). Laurdan dye binds to membranes and emits light at 440 nm in gel-phase membranes (packed) and at 490 nm in liquid-phase membranes (loose) (Fig. 5a). The *AdipoR2* KO cells showed a significantly higher generalized polarization (GP) index compared to WT cells in both spermatocytes and round spermatids (Fig. 5b, c), suggesting that the membrane became more packed and thus less fluid in the mutant cells. Moreover, bar-shaped solid-like membranous structures were frequently observed in the cytoplasm of *AdipoR2* KO cells (Fig. 5b, asterisks), which likely correspond to the rigidified endoplasmic reticulum membranes reported in *AdipoR1* and *AdipoR2* double KO mouse embryonic fibroblasts[26].

A unique event in meiosis is the attachment of telomeres to the NE. Telomere attachment to the NE is mediated by a meiosis-specific protein complex, TERB1-TERB2-MAJIN, which recruits the SUN1-KASH5 transmembrane complex for the movement of telomeres (Fig. 5d)[18,19,31,32]. Telomere movement along the NE facilitates homologous pairing, synapsis, and recombination and thus is crucial for faithful meiotic progression. Interestingly, the staining of the telomere marker TRF1 in *AdipoR2* KO pachytene spermatocytes showed a significant increase in internalized telomeres dislocated from the nuclear periphery (Fig. 5e, asterisks). Notably, these internalized telomeres coincided with the internalization of Lamin B, which is a marker for the nuclear lamina normally attached to the inner NE (Fig. 5e). The same

phenotypes were observed by the staining of TERB1 and TERB2 (Supplementary Fig. 7a, b). Further, the integral inner NE proteins, MAJIN, SUN1, and the outer NE protein, KASH5, which accumulate at the telomere attachment sites, were also internalized (Fig. 5e), suggesting that telomeres are internalized together with the inner and outer NE components.

The TERB1-TERB2-MAJIN complex is responsible for the formation of the telomere-NE fusion structure, known as the telomere attachment plate, as seen by TEM[19]. The TEM investigation of *AdipoR2* KO spermatocytes showed that the formation of the telomere attachment plate was intact, which was in contrast to the *Terb2* KO spermatocytes where the formation of the telomere attachment plate was completely abrogated (Fig. 5f). However, the TEM analysis still showed characteristic NE defects in *AdipoR2* KO spermatocytes involving the invagination of the NE toward the inside of nucleoplasm (Fig. 5g, left). The NE invagination appeared like a membranous bubble inside the nucleus when seen in cross-section (Fig. 5g, right). Telomeres were frequently found attached to the membranous bubble (Fig. 5h and Supplementary Fig. 7c). The membranous bubble enclosed the mitochondria, supporting the notion that it is a cytoplasmic compartment invaginated into the nucleus (Fig. 5h). Taken together these results suggest that the stiffening of the nuclear membrane in *AdipoR2* KO spermatocytes leads to the NE invagination, which disturb the nuclear peripheral distribution of telomeres in meiosis. It is likely that the dynein-dependent forces applied to telomeres, which normally induce telomere movement within the fluid membrane in WT, instead lead to NE curvature and subsequent invagination in cases where telomeres are tightly embedded within the rigidified membrane.

## Destabilization of the intercellular bridge disrupts the germ cell syncytium

Another distinctive characteristic of the cellular membrane in germ cells is the formation of intercellular bridges between synchronously developing germ cells[5]. The intercellular bridge is visualized by the staining of the integral component TEX14[7]. The intercellular bridge matures during germ cell development, and the diameter of the TEX14 ring became larger from spermatocytes to round spermatids (Fig. 6a). In *AdipoR2* KO testis the staining of TEX14 showed clear abnormalities. First, the diameter of the TEX14 ring was abnormally larger in round spermatids compared to WT spermatocytes (Fig. 6a). Further, the TEX14 signals sometimes appeared as elongated linear structures surrounding the multinucleated giant round spermatids, and the individual spermatid nuclei were no longer connected by the TEX14 ring (Fig. 6b and Supplementary Movies 1–3). These results suggest that the integrity of the intercellular bridge was significantly impaired in *AdipoR2* KO cells leading to the defects in the formation of germ cell

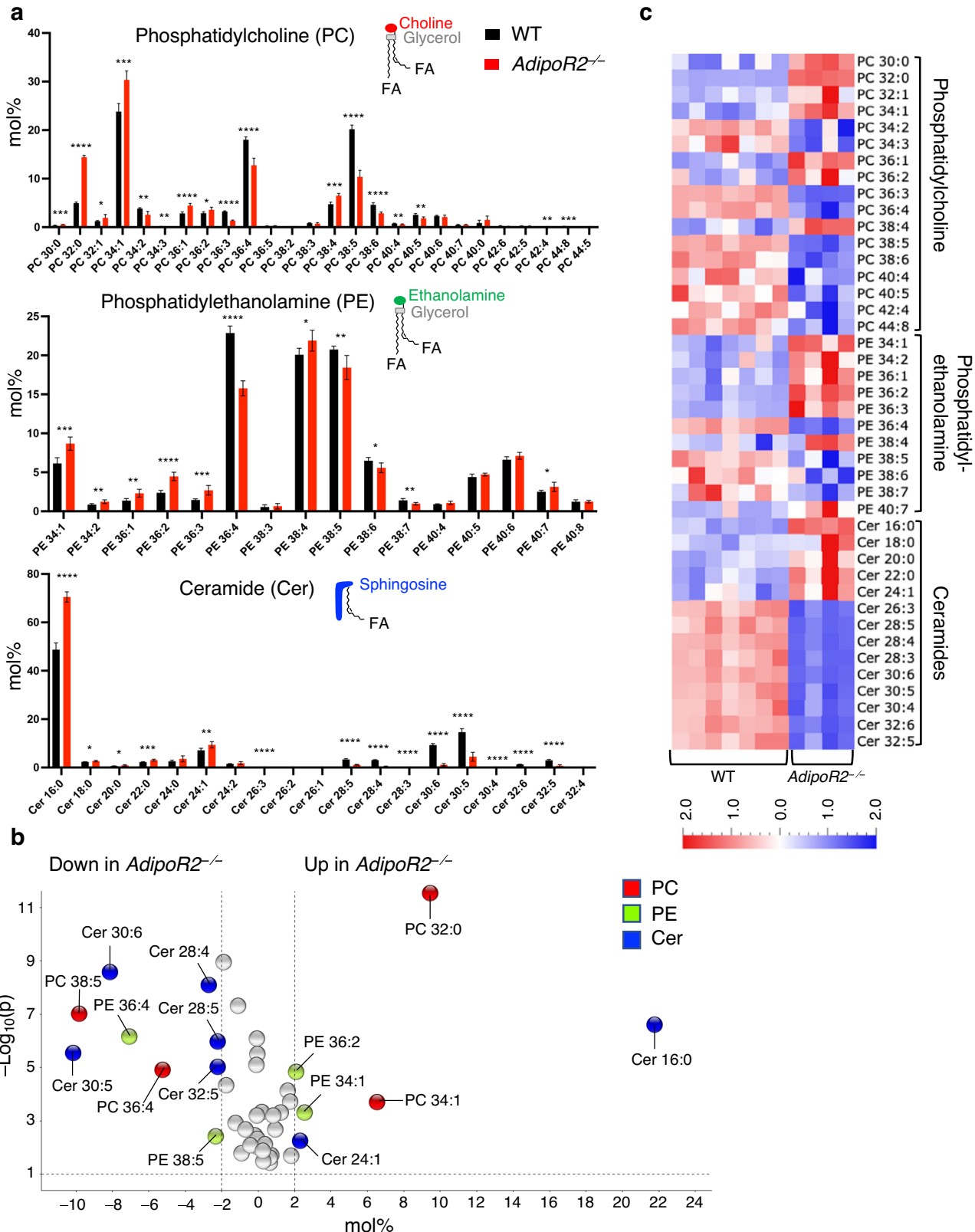

**Fig. 3 | The depletion of VLC-PUFA in the *AdipoR2*$^{-/-}$ testis lipidome. a** The amounts of each indicated lipid species in PD28 testes from WT and *AdipoR2*$^{-/-}$ male mice are shown as mol%. $n = 7$ and $n = 4$ biological replicates were used for WT and *AdipoR2*$^{-/-}$, respectively. Mean values with SD are shown. Two-tailed *t*-tests. \**p* > 0.05, \*\**p* > 0.01, \*\*\**p* > 0.001, \*\*\*\**p* > 0.0001. The absolute amounts are presented in Supplementary Fig. 4. **b**, **c** Volcano plots (**b**) and heat maps (**c**) of the lipid species in PD28 testes from WT and *AdipoR2*$^{-/-}$ male mice. Only lipid species that showed significance with $p < 0.05$ were included. *p*-values were calculated by a two-sided *t*-test. The means for each lipid species were adjusted to 0 and the variance was adjusted to 1 in the heat maps. For PCs and PEs, it was not possible to determine the actual FA composition because of the ultra-long-chain species present. The data for PCs and PEs are therefore presented as whole phospholipids. Source data are provided as a Source Data file.

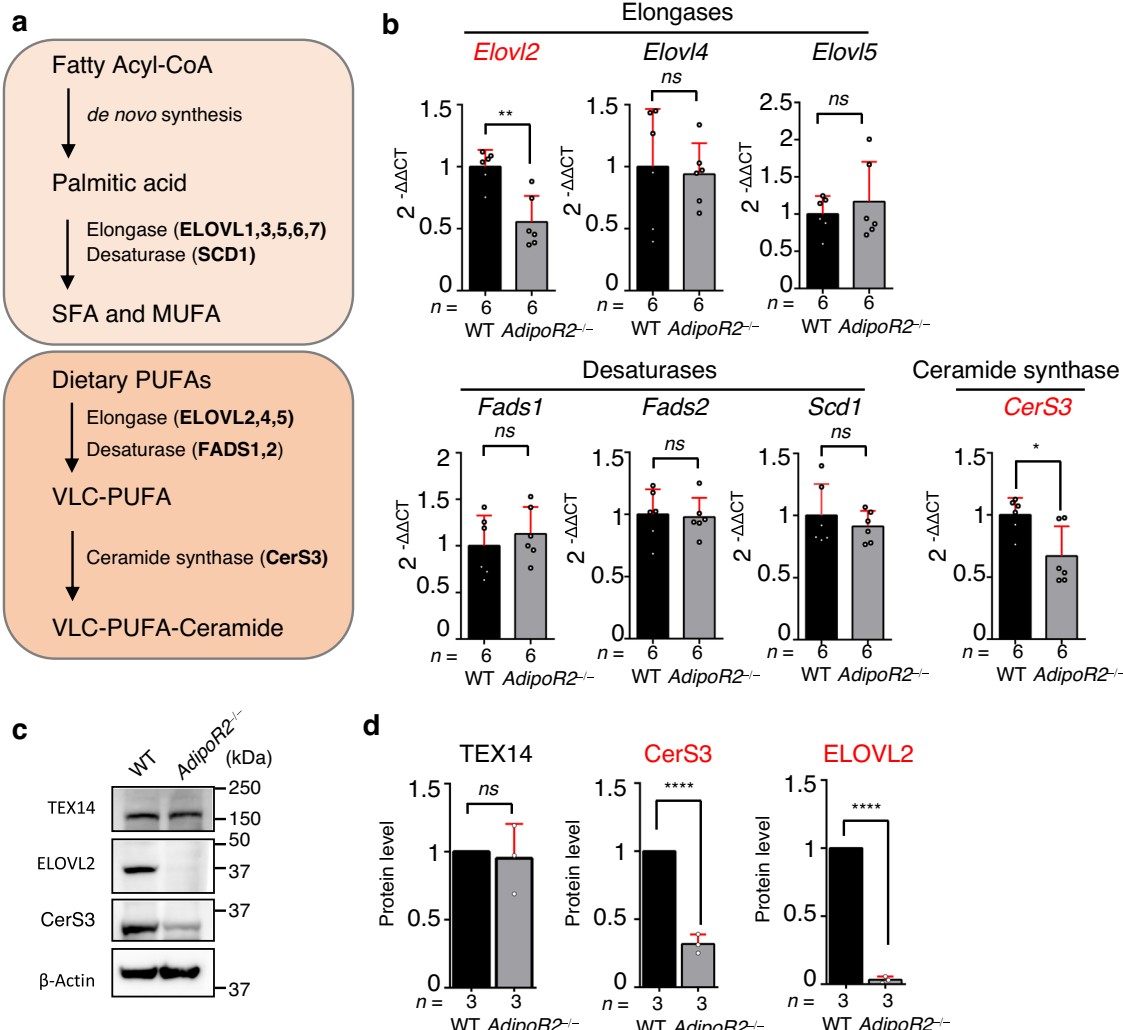

**Fig. 4 | AdipoR2 regulates ELOVL2 both transcriptionally and post-transcriptionally. a** Simplified model of fatty acid biosynthesis, including key enzymes involved in each step. **b** Gene expression changes normalized to *GAPDH* in testes from WT and *AdipoR2⁻/⁻* male mice analyzed by qPCR. *n* = 6 testes pooled from three mice for each genotype. Mean values with SD are shown. Two-tailed *t*-tests. ns: not significant. **p* = 0.0013, ***p* = 0.014. **c** Immunoblots of mouse testis extracts from WT and *AdipoR2⁻/⁻* males. **d** The quantification of immunoblots normalized to the mean value for WT. The band intensities were normalized to β-actin. *n* = 3 tests per sample. Mean values with SD are shown. Two-tailed *t*-tests. ns: not significant. *****p* < 0.0001. Source data are provided as a Source Data file.

syncytia. Similar defects in TEX14 integrity were reported in germ cell-specific conditional knockouts of *CerS3* and *Gcs* (a gene encoding a glucosylceramide synthase responsible for the addition of glycan moieties to sphingolipids)[9], suggesting that the defects in the intercellular bridge integrity are hallmark issues caused by abnormal sphingolipid and glycosphingolipid composition in male germ cells.

Male germ cells are orderly arranged within the seminiferous tubules (Fig. 6c). To test if the defects in intercellular bridge formation affect this tubular structuring, we investigated the distribution of different germ cell types in *AdipoR2* KO seminiferous tubules. In WT cases, spermatocytes localized to the periphery of the seminiferous tubule, while spermatids localized to the internal lumina, and thus spermatocytes and spermatids never coexisted in the same niche within a seminiferous tubule. However, in *AdipoR2* KO seminiferous tubules spermatocytes were frequently localized in the internal lumen and mixed with round spermatids in the same niche (Fig. 6d). The immunostaining of SYCP3 confirmed that SYCP3-positive spermatocytes were indeed localized in the luminal region of seminiferous tubules and were juxtaposed with round spermatids in *AdipoR2* KO seminiferous tubules (Fig. 6e).

Sertoli cells create the testicular cell niche by forming the blood-testis barrier (BTB), which separates the seminiferous tubule into a basal compartment and an endoluminal compartment[33]. The proper composition of lipid species in the testis is essential for the integrity of the BTB, as demonstrated by its disruption in ether lipid-deficient mice[34]. To exclude the possibility that the abnormal distribution of germ cells within the *AdipoR2* seminiferous tubule is a consequence of similar Sertoli cell defects, we tested the integrity of the BTB using a biotin tracer assay. In WT mice, the biotin tracer was observed in the basal compartment but was excluded from the endoluminal compartment (Fig. 6f, left). Cadmium chloride disrupted the BTB, leading to the leakage of the biotin tracer into the endoluminal compartment (Fig. 6f, right). In the *AdipoR2* KO seminiferous tubule, however, the biotin tracer was restricted to the basal compartment and did not spread to the endoluminal compartment, which was similar to WT (Fig. 6f, middle). These data suggest that the integrity of the BTB is intact in *AdipoR2* KO seminiferous tubules. Thus, we conclude that the orderly arrangement of cell types within the seminiferous tubules was impaired following the destabilization of intercellular bridges but not the BTB in *AdipoR2* KO seminiferous tubules (Fig. 6f).

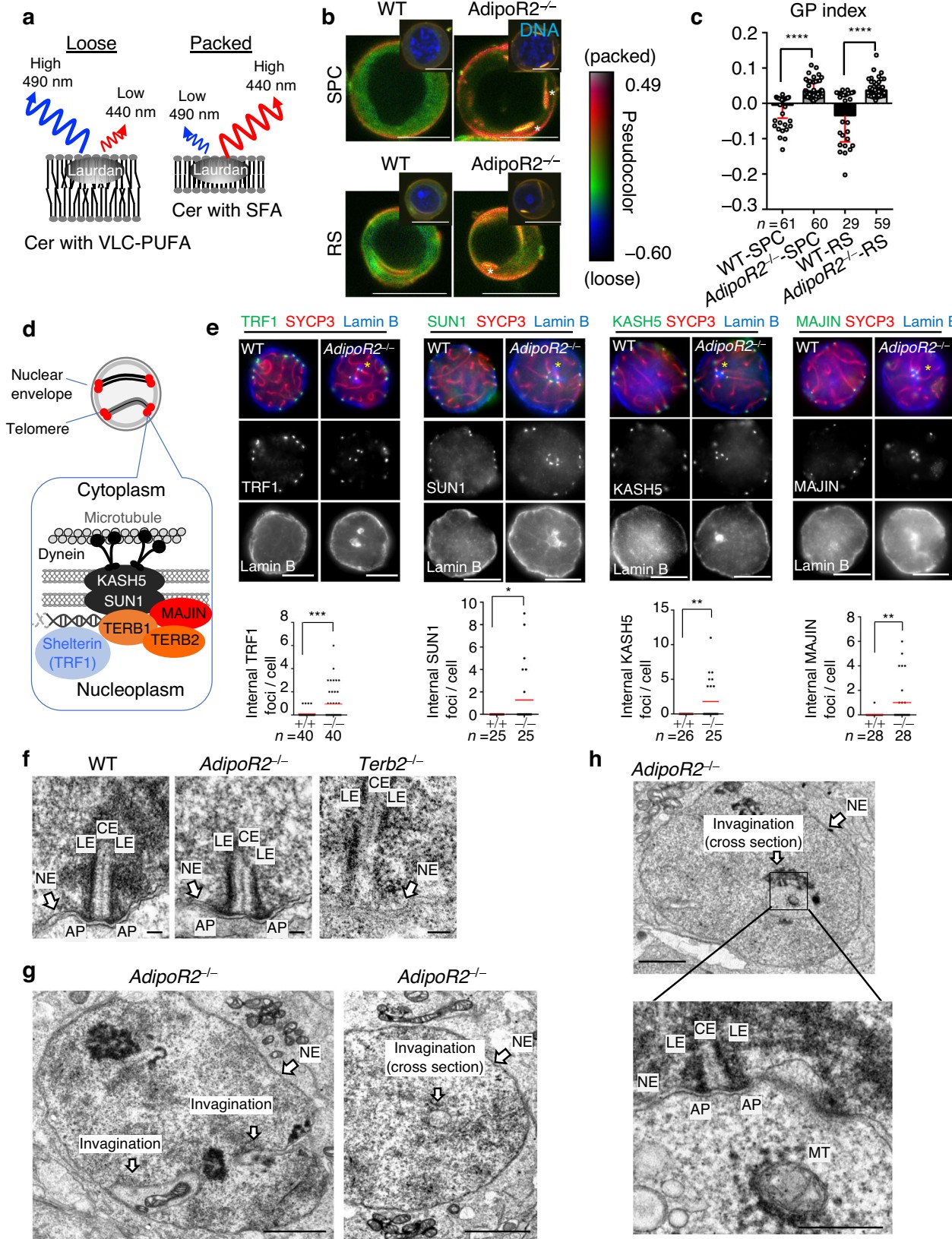

## Discussion

AdipoR1/2 were identified as the receptors for the adipocyte-secreted protein adiponectin encoded by the *Adipoq* gene[23], but later studies demonstrated that they are evolutionarily conserved regulators of membrane fluidity[24,35,36]. In this study, we have shown that AdipoR2 is necessary for fluidizing the cell membrane in male germ cells and thus ensuring proper meiosis progression and male fertility (Fig. 7). *AdipoR2* deficiency impaired the nuclear peripheral distribution of meiotic telomeres, homologous synapsis, and recombination in spermatocytes, providing evidence linking membrane fluidity defects with specific chromosomal anomalies in meiosis. Of note, *Adipoq* KO mice were reported to be fertile[37], suggesting that AdipoR2's role in the

**Fig. 5 | A rigidified membrane impairs the telomere distribution in meiosis.**
**a** Schematic of the Laurdan dye method. **b** Representative pseudocolor images of spermatocytes (SPC) and round spermatids (RS) from WT and *AdipoR2⁻/⁻* male mice showing the Laurdan dye GP index at each pixel position. Note the pronounced rigidification of *AdipoR2⁻/⁻* cells as shown by red pixels and the presence of long and packed structures (indicated by asterisks). Scale bars: 10 µm. **c** The average GP index with SD. Two-tailed *t*-tests. ****$p < 0.0001$. **d** Schematic of the molecular structure of meiotic telomeres attached to the NE. **e** Immunostaining of pachytene spermatocytes from WT and *AdipoR2⁻/⁻* male mice. Equatorial sections are shown. The graph shows the mean internal telomeric foci number. *n* shows the number of spermatocytes pooled from three mice for each genotype. Two-tailed *t*-tests. TRF1

$p = 0.0050$, SUN1 $p = 0.018$, KASH5 $p = 0.0031$, MAJIN $p = 0.0068$. Scale bars: 5 µm. Source data are provided as a Source Data file. **f** TEM images of telomere attachment plates near the NE in spermatocytes from WT, *AdipoR2⁻/⁻*, and *Terb2⁻/⁻* male mice. LE: lateral element. CE: central element. AP: attachment plate. NE: nuclear envelope. Scale bars: 100 nm. Experiments were repeated in a total of 3 mice. **g, h** TEM images of entire spermatocytes from *AdipoR2⁻/⁻* male mice. The invaginations of the NE are indicated. Scale bars: 2 µm (500 nm in the magnified panel). The telomere attachment at the invaginated NE is magnified in **h**. LE lateral element, CE central element, AP attachment plate, NE nuclear envelope. MT mitochondria. Another example showing the telomere attachment to the invaginated NE is provided in Supplementary Fig. 7c. Experiments were repeated in a total of 3 mice.

testis is likely to be independent of adiponectin binding, consistent with the reported adiponectin-independent roles of AdipoRs in *C. elegans* (where no adiponectin homolog is conserved) and in cultured mammalian cells[25].

AdipoR2 exhibits ceramidase activity, hydrolyzing ceramides to produce sphingosine and free FAs[38]. The resulting sphingosine can be phosphorylated to produce the signaling molecule sphingosine-1-phosphate (S1P). In *C. elegans* and cultured mammalian cells, S1P is a critical downstream effector of AdipoR2 that activates specific transcription factors and promotes the transcription of FA desaturases, particularly the stearoyl-COA desaturase (SCD)[26]. Our study found that *AdipoR2* KO did not affect the mRNA expression of SCD1 (the murine homolog of human SCD) in germ cells. Instead, we found that the upregulation of CerS3 and ELOVL2 at both the transcriptional and post-transcriptional levels is the key downstream event regulated by testicular AdipoR2. In particular, the expression of the ELOVL2 protein was almost completely abolished in *AdipoR2* KO testis. AdipoR2 and its *C. elegans* homolog PAQR-2 directly interact with an FA elongation complex[39], suggesting that this physical interaction may contribute to the stabilization of ELOVL2 in the testis. Overall, our findings imply that AdipoR2 ensures membrane fluidity homeostasis by controlling a comprehensive network involved in lipid metabolism, including gene transcription and protein stability, beyond the previously identified SCD-dependent pathway.

The synthesis of sphingolipids and the VLC-PUFA chain in testes depends on *CerS3* and *Elovl2*, respectively, and the deletion of these genes causes cell death in spermatocytes[9,10]. In *AdipoR2* KO testis, the levels of ceramides were unchanged, but VLC-PUFA chains were largely absent from ceramides due to a significant reduction in ELOVL2 protein expression. Previous studies didn't delve into the specific meiotic defects caused by abnormalities in the lipidome in testes. The present work identifies multiple meiotic chromosomal defects arising from the loss of VLC-PUFA. Some chromosomal defects may be direct consequences of membrane rigidification. For example, the defects in telomere attachment to the NE correlated with altered NE morphology, such as pronounced invaginations. In *AdipoR2* KO spermatocytes with a highly-rigid NE lacking VLC-PUFA, the cytoplasmic dynein-dependent movement forces on telomeres might induce NE curvature, contrasting with the movements of telomeres within the fluid membrane observed in the wild-type scenario. Chromosome movements driven by the NE-attached telomeres are required for faithful homologous pairing and recombination. The defective NE properties and the telomere distribution can thus explain the observed homologous synapsis and recombination errors in *AdipoR2* KO spermatocytes, similar to what has been reported in a number of different KO mice defective in telomere-driven chromosome movements[18,19,31,40,41]. The necessity of unsaturated FAs for NE architecture appears to be evolutionarily conserved, as observed in recent findings highlighting their importance for the NE structure and function in yeast cells[42]. Alternatively, the

defects in the formation of the intercellular bridges could be one of the indirect causes of the observed meiotic errors in *AdipoR2* KO spermatocytes. The *Tex14* KO mice showed meiotic arrest in prophase I[7], suggesting that defects in the intercellular bridges alone without lipidome defects are sufficient to cause prophase I arrest. In summary, the meiotic defects observed in *AdipoR2* KO testes are likely to result from a complex interplay, with direct consequences arising from membrane rigidification and indirect effects stemming from the disruption of intercellular bridges.

In addition to testes, VLC-PUFAs are also abundant in the retina and play crucial roles in retinal biology and photoreceptor renewal[43,44], and both *Elovl2* and *AdipoR1* are reported to be essential for normal retinal functions[44,45]. In the retina, ELOVL2-dependent VLC-PUFA synthesis is progressively attenuated with aging due to an age-related increase in *Elvol2* gene methylation[44,46,47]. Similarly, *AdipoR1* deficiency showed specific defects in the retina, where PCs containing VLC-PUFAs are significantly decreased leading to progressive photoreceptor degeneration[48]. A recent study reported a molecular link showing that AdipoR1 upregulates *Elovl2* at the transcriptional level in the retina[49], paralleling our discovery of AdipoR2's roles in the testis. Thus, the AdipoR1/2-ELOVL2 functional axis as the master regulator of VLC-PUFA synthesis is likely to be conserved beyond the specific tissue contexts studied here.

## Methods
### Mouse
Mice were congenic with the C57BL/6J background. Knockout mice for *AdipoR2* and *Terb2* were reported earlier[19,50]. All animal experiments were approved by the Regional Ethics Committee of Gothenburg, governed by the Swedish Board of Agriculture (#1316/18).

### Reverse transcription PCR
Total RNA was isolated from tissues using the RNeasy Mini kit (Qiagen). cDNAs were generated using the iScript reverse transcription super mix (Bio-Rad), and PCR amplification was performed using standard DNA polymerase. The primers designed for this study are shown in Supplementary Table1.

### Histological analysis
Testes and epididymides were fixed in Bouin's fixative for 24 h at room temperature and embedded into paraffin blocks. Slices of 8 µm thickness were stained with hematoxylin and eosin. TUNEL analysis was carried out with an ApopTag Plus In Situ Apoptosis Fluorescein Detection Kit (S 7111; Millipore).

### Antibodies
The following antibodies were used: rabbit antibodies against AdipoR2 (1:800 for WB)[50], TEX14 (1:1000 for WB, 1:200 for IF, Abcam; Ab41733, GR109649-1), SYCP1 (1:5000 for IF, Abcam; Ab15090, GR3184119-1), γH2AX (1:3000 for IF, Abcam; Ab11174, GR294890-8), RAD51 (1:500 for IF, Thermo Fisher Scientific; PA5-27195, UI2840658J), ELOVL2 (1:1000

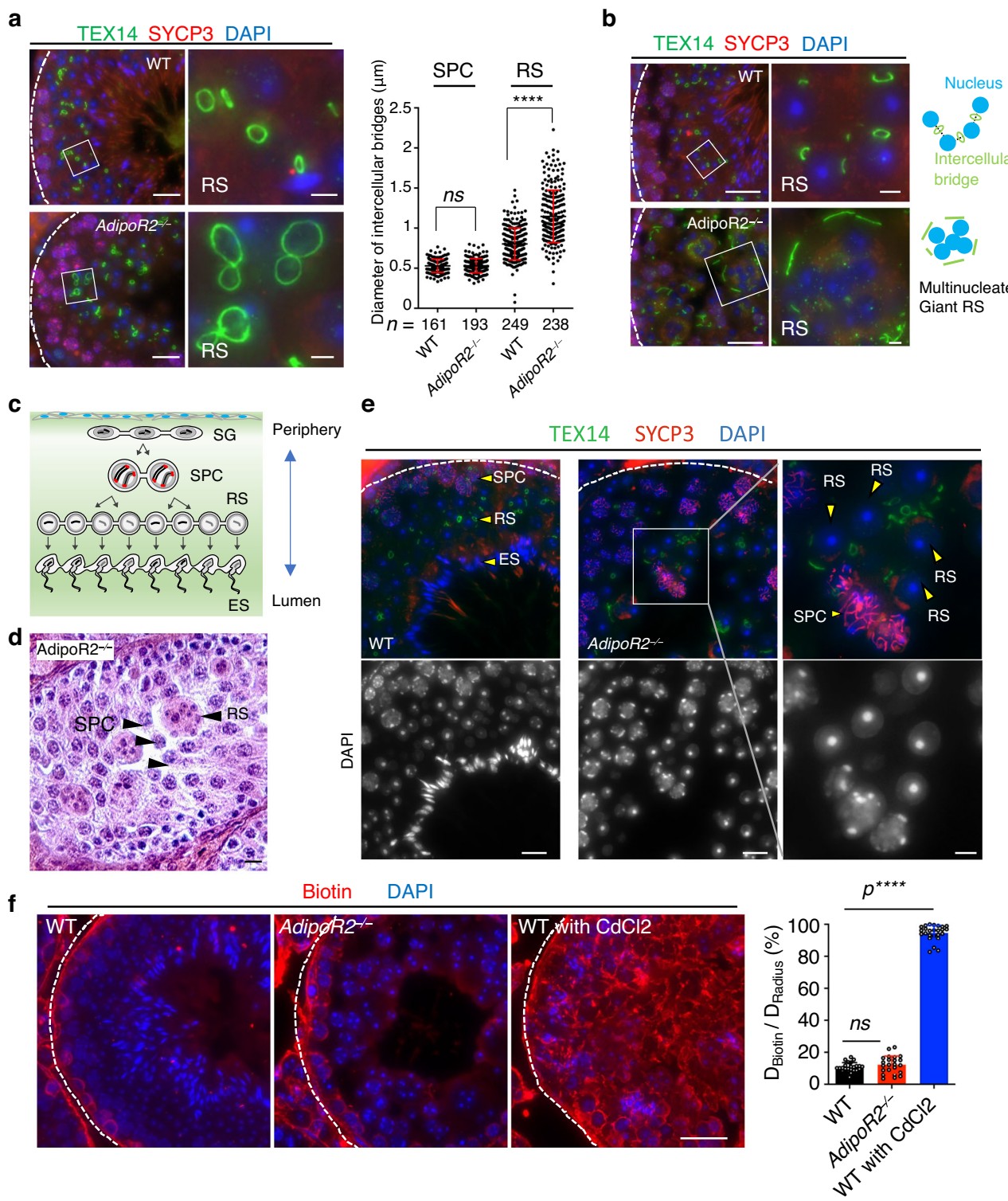

for WB, Abcam; Ab176327, GR139744-4), CerS3(1:1000 for WB, 1:100 for IF)[51], SUN1 (1:1000 for WB, 1:100 for IF, Abcam; Ab103021, 1014380-1), KASH5 (1:1000 for IF, Hiroki Shibuya lab), MAJIN (1:1000 for IF)[19], TERB1 (1:1000 for IF)[18], and TERB2 (1:1000 for IF)[19]; rat antibody against RPA2 (1:200 for IF, Cell Signaling Technology; 2208 S, 3); mouse antibodies against TRF1(1:1000 for IF)[18], MLH1 (1:50 for IF, BD Biosciences; 51-1327GR, 4136717), and β-Actin (1:2000 for WB, Sigma; A2228-200UL, 067M4856V); goat antibody against Lamin B (1:100 for IF, Santa Cruz Biotechnology; sc-6216, F1715); and chicken antibody against SYCP3 (1:5000 for IF)[52].

**Immunostaining of spermatocytes**
Testis cell suspensions were prepared and washed in PBS, centrifuged, and resuspended in hypotonic buffer (30 mM Tris (pH 7.5), 17 mM trisodium citrate, 5 mM EDTA, and 50 mM sucrose) followed by centrifugation and resuspension in 100 mM sucrose. The cell suspensions were placed on slides in the same volume of fixation buffer (1% paraformaldehyde and 0.1% Triton X-100), fixed for 3 h at room temperature, and air dried. For immunostaining, the slides were incubated with primary antibodies in PBS containing 5% BSA for 2 h and then with Alexa Fluor 488-, 594-, or

**Fig. 6 | Defects in the formation of intercellular bridges and germ cell syncytium. a** Immunostaining of testis sections from 8-week-old WT and *AdipoR2⁻/⁻* males. Round spermatids (RS) are magnified. Scale bars: 10 µm (2 µm in the magnified panel). The graph shows the quantification of the diameter of the intercellular bridge stained by TEX14 in spermatocytes (SPC) and RS. Mean values with SD are shown. *n* shows the number of intercellular bridges pooled from three mice for each genotype. Two-tailed *t*-tests. ns: not significant. ****p < 0.0001. **b** Immunostaining of testis sections from 8-week-old WT and *AdipoR2⁻/⁻* males. Round spermatids (RS) are magnified and shown schematically. Scale bars: 10 µm (2 µm in the magnified panel). **c** Schematic of spermatogenesis progression within the seminiferous tubule. SG: spermatogonia. SPC: spermatocyte. RS: round spermatid. ES: elongated spermatid. **d** Testis sections from 8-week-old *AdipoR2⁻/⁻* males stained with hematoxylin and eosin. The arrowheads indicate internalized spermatocytes

(SPC) juxtaposed with round spermatids (RS). Scale bar: 10 µm. Experiments were repeated in a total of 3 mice. **e** Immunostaining of testis sections from 8-week-old WT and *AdipoR2⁻/⁻* males. SPC: spermatocyte. RS: round spermatid. ES: elongated spermatid. Scale bars: 10 µm (2 µm in the magnified panel). Experiments were repeated in a total of 3 mice. **f** The restriction of the biotin tracer to the basal compartment in both WT and *AdipoR2⁻/⁻* male testes. As a control, mice were treated with CdCl2. Scale bars: 15 µm. The permeability of the BTB was quantified by the ratio of diffusion distance of the biotin tracer (DBiotin) and the radius of the corresponding tubule (DRadius). Data are shown as the mean with SD of 24, 23, and 25 tubules randomly selected from two mice for WT, *AdipoR2⁻/⁻*, and WT with CdCl2, respectively. One-way ANOVA with Dunnett's multiple comparisons test. ns: not significant, ****p < 0.0001. Source data are provided as a Source Data file.

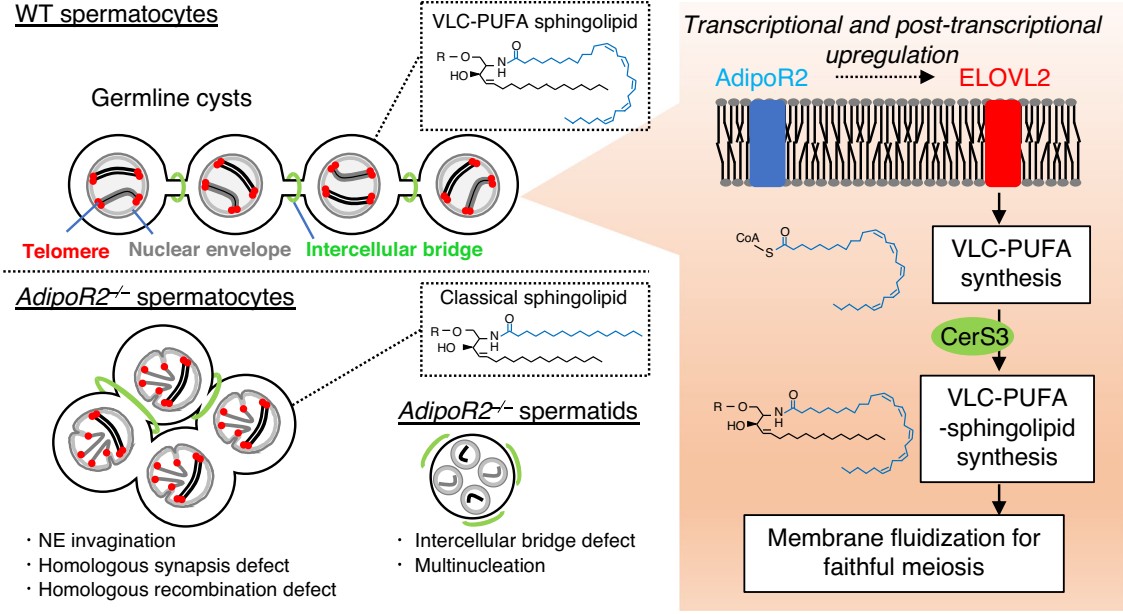

**Fig. 7 | Proposed model.** AdipoR2 regulates the testis-specific synthesis of VLC-PUFA sphingolipid by controlling lipid biosynthesis enzymes such as ELOVL2 and CerS3, ensuring the fluidity of the cellular membrane. When the production of VLC-PUFAs is disrupted, it leads to the stiffening of the cellular membrane and nuclear envelope, causing defects in meiotic chromosomal dynamics and impacting germline cyst formation.

647-conjugated secondary antibodies (1:1000 dilution, Invitrogen) for 1 h at room temperature. The slides were washed with PBS and mounted with VECTASHIELD medium with DAPI (Vector Laboratories). Signal intensities were measured using the Soft-Worx Data Inspector tool. The background signals were measured in the same way and subtracted.

## Lipidomics

Mouse testes were removed and immediately frozen on dry ice, and tissues were stored at −80 °C until analysis. For lipid extraction, tissues were extracted using the BUME method according to previous work[53]. Internal standards were added during lipid extraction. Phospholipids and sphingomyelins were measured using direct infusion mass spectrometry. For phospholipids, a part of the lipid extracts was evaporated and reconstituted in chloroform:methanol [1:2] with 5 mM ammonium acetate. This solution was infused directly (shotgun approach) into a QTRAP 5500 mass spectrometer (Sciex) equipped with a TriVersa NanoMate (Advion Bioscience) as described previously[54]. A similar procedure was used for sphingomyelin. However, prior to analysis the lipid extracts were evaporated and exposed to alkaline hydrolysis (0.1 M KOH in methanol for 60 minutes at room temperature) in order to

remove the interfering phospholipids. For phosphatidylcholines and sphingomyelins, mass spectra were obtained in precursor ion scanning mode using the phosphocholine headgroup (*m/z* 184.1) as the fragment ion. For phosphatidylethanolamines, neutral loss scanning of m/z 141.1 was used[55,56]. Ceramides were measured using ultra-performance liquid chromatography coupled to tandem mass spectrometry as previously described[57]. In brief, the separation of ceramides species was performed using a Waters BEH C8 column (2.1 × 100 mm with 1.7 µm particles) with water, acetonitrile, and isopropanol as mobile phases. The detection was made in positive MRM (targeted) mode using *m/z* 264 as a fragment ion. The lipidomics data were evaluated using the LipidView and MultiQuant software (Sciex), and Qlucore Omics Explorer software was used for the multivariant analysis. The complete lipid composition data are provided in the Source data.

## Measurement of membrane packing

Live cells were stained with Laurdan dye (6-dodecanoyl-2-dimethylaminonaphthalene) (Invitrogen) at 15 µM for 1 h. Images were acquired with an LSM880 confocal microscope (Zeiss) equipped with a live cell chamber (set at 33 °C and 5% CO2) and ZEN software

with a 40× water-immersion objective. Briefly, cells were excited with a 405 nm laser and the emission was recorded between 410 nm and 461 nm (ordered phase) and between 470 nm and 530 nm (disordered phase). Images were acquired with 16-bit image depth and 1024 × 1024-pixel resolution using a pixel dwell of ~1.02 µs. The GP index was calculated using ImageJ version 1.47 software following published guidelines[58]. SYTO59 (5 mM in DMSO) (Invitrogen) was added together with the Laurdan dye to stain the nuclei and allow distinguishing between cell types. SYTO59 was excited with a 561 nm laser, and the emission was recorded between 570 nm and 758 nm.

## Biotin tracer assay

WT and *AdipoR2* KO mice were anesthetized, and 20 µl EZ-Link Sulfo-NHS-LC-Biotin (10 mg/ml in PBS containing 1 mM CaCl2) (Thermo Fisher Scientific) was injected into the testis interstitium. Mice were euthanatized after 40 minutes. The testes were removed and fixed in 4% formaldehyde overnight, and 5 µm sections were obtained and stained with FITC-conjugated streptavidin (Invitrogen) and DAPI. For quantification, the distance traveled by biotin ($D_{Biotin}$) from the basement membrane in a seminiferous tubule versus the radius of the tubule ($D_{Radius}$) was calculated. For oval tubules, the average of the longest distance and the shortest distance from the basement membrane was used. As the positive control, WT mice were treated with daily doses of CdCl2 (5 mg/kg, i.p.) (Sigma) for 3 consecutive days before the BTB assay.

## qPCR

Total RNA was extracted from both WT and *AdipoR2* KO testis biopsies using the RNeasy Mini Kit according to the manufacturer's protocol (Qiagen) and quantified by a NanoDrop spectrophotometer (ND-1000; Thermo Fisher). cDNA was synthesized using the SuperScript III First-Strand Synthesis System (Invitrogen), and the qPCR experiment was performed with a CFX Connect Real-Time PCR Detection System (BioRad) using the iTaq Universal SYBR Green Supermix qPCR system (BioRad) and standard primers. The relative expression of each gene was measured by the delta-Ct method. The primers designed for this study are shown in Supplementary Table1.

## TEM

For TEM, testis tissue samples were prepared and fixed as previously described[59]. Small testis tissue pieces were fixed at 4 °C in cacodylate-buffered glutaraldehyde solution (2.5% glutaraldehyde, 50 mM KCl, 2.5 mM MgCl, 50 mM cacodylate; pH 7.2) for 1 h and subsequently washed five times for 3 min each in fresh cacodylate buffer (50 mM cacodylate, pH 7.2). The samples were then incubated in 2% osmium tetroxide in 50 mM cacodylate for 2 h at 4 °C washed three times in distilled H$_2$O and contrasted overnight at 4 °C with 0.5% uranyl acetate (in distilled H$_2$O). The tissue pieces were dehydrated in an increasing ethanol series and incubated three times in propylene oxide for 30 min each time at room temperature. Finally, the samples were embedded in epon. Ultrathin sections (65 nm to 100 nm) were transferred to Formvar-coated copper grids and treated with fresh 2% uranyl acetate solution in distilled H$_2$O for 20 min at room temperature. After washing in distilled H$_2$O, testis sections were counterstained in Reynold's lead citrate for 10 min, washed in distilled H$_2$O, and dried at room temperature. Samples were analyzed with a JEOL JEM-1400 Flash Scanning Transmission Electron Microscope (Jeol, Eching, Germany) operated at 120 kV. For the analysis of *Terb2* KO samples, some minor changes in the fixation/embedding protocol were performed as previously described[19].

## Reporting summary

Further information on research design is available in the Nature Portfolio Reporting Summary linked to this article.

## Data availability

All data are available in the main text or the supplementary materials. All other data supporting the findings of this study are available from the corresponding author upon request. Source data are provided in this paper.

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

## Acknowledgements

We thank Roger Sandhoff (German Cancer Research Center) for providing the CerS3 antibody. The Jeol JEM-1400 STEM was funded by the Deutsche Forschungsgemeinschaft (DFG, German Research Foundation)—426173797 (INST 93/1003-1 FUGG). This work was supported by the German Research Foundation Al 1090/4-2 (M.A), the European Research Council StG-801659 (H.S.), the Swedish Research Council 2018-03426 (H.S.), and the Knut och Alice Wallensbergs Stiftelse KAW2019.0180 (H.S.).

## Author contributions

J.Z., M.R., N.S., R.D. and M.B. performed the mouse experiments and analyzed the data; P.O.B. and M. Henricsson performed the lipidomics analysis with supervision from J.B.; M.H., M.A., and A.H. performed the TEM analysis; M.P. and H.S. supervised the project; and H.S. wrote the manuscript.

## Funding

## Competing interests

The authors declare no competing interests.
