## [Peer Review File · Nature Communications]

Regulation of meiotic telomere dynamics through membrane fluidity promoted by AdipoR2-ELOVL2Reviewers' Comments:

Reviewer #1:

Remarks to the Author:

All somatic cells express in their membranes sphingolipids (ceramides, sphingomyelins and glycosphingolipids), which contain saturated long and saturated or monounsaturated very long acyl chains. In contrast male fertility requires the expression of very unique sphingolipids containing even longer very long and simultaneously also polyunsaturated acyl chains (VLC-PUFA sphingolipids), which are upregulated in spermatocytes during spermatogenesis. Their requirement for spermatogenesis had been proven using knockout mice lacking enzymes responsible for their synthesis (Elovl2, CerS3, Ugcg, B4galnt1). These lipid structures are also found in human testis, indicating corresponding functions. Loss of these sphingolipids leads to an arrest in spermatogenesis, which was shown to go along with formation of multinucleated giant cells and loss of proper intercellular bridges between clonal spermatids. However, detailed mechanisms on the molecular function of these unique sphingolipids still remain elusive as well as regulation of their synthesis.

With this manuscript, the authors add a major puzzle piece to the understanding of the regulation of VLC-PUFA sphingolipids synthesis and give insight into further fundamental cellular processes, impaired by the absence of these unique sphingolipids. Therefore, this manuscript should be of great interest to the readership of Nature Communications as it exemplifies, how specific cellular processes are supported by a correspondingly adjusted lipid membrane composition.

Major comments

To support the conclusions of the authors, it would be very helpful to show by immunofluorescence of wild type testis, which cell types and differentiation stages express AdipoR2 to what kind of degree. Specifically the subcellular localization of AdipoR2 in spermatocytes would be very informative, i.e. is it plasma membrane staining or is it ER/nuclear envelope staining, or both.

During attachment of telomeres to the nuclear envelope: is there really an internalization of telomeres together with the inner and outer NE components, or is this always an unusual invagination of the nuclear membranes as depicted in figures 5f-h? This should be clarified.

Furthermore, can such invaginations be explained by stiffening of the lipid membrane? Wouldn't higher curvatures, needed for such invaginations, be in favor of more fluid membranes?

In this context a higher magnification and resolution of TEM pictures should be provided, in addition.

The interpretation of the authors implies that the VLC-PUFA sphingolipids are constituents of the nuclear membrane. Can the authors add any data supporting this assumption?

Finding spermatocytes more luminal than round spermatids indicates defects in normal tubular structuring. However, does that really proof an impaired synchronicity within one clone of maturing germ cells? For this, I guess, it would be necessary to demonstrate that these spermatocytes and round spermatids (Fig. 6e) are within one syncytium, i.e. not separated by cell membranes, which would require analysis with electron microscopy. It may very well be, that after formation of multinucleated giant cells, the spermatids included continue to differentiate in a synchronous fashion until a certain point, i.e. initiation of acrosome formation.

Minor comments

The authors mixed up all links for figures in the main text with the numbering of figures and extended figures. Some extended figures appear as main figures. This made reading quite difficult and has to be corrected. I do not understand how 13 authors could have missed to see this. Extended Figure 2e, as mentioned in the main text, does not exist at all.

Page 4 lines 88/89: The statement is not proven by Fig. 1d as the resolution of Fig. 1d does not allow the reader to judge if MGs are filled with nuclei of spermatocytes or spermatids and not of which stage spermatocytes could be.

Page 6 lines 136/137: I consider a complete loss of a lipid more significant than an increase of a lipid by 22 mol%. Moreover, the loss of ceramides with VLC-PUFAs automatically has to be compensated in an increase of another ceramide species as the representation of mol% will always add up to 100%. Hence: Presentation of lipid data in mol% may be misleading and does not allow to conclude if a subspecies is down or up regulated. Therefore absolute or exact lipid concentrations have to be plotted, which allow the reader to judge if, i.e. a ceramide with palmitic acid is really upregulated or its mol% simply increases, because ceramides with PUFAs are downregulated. If this is not possible, the relative amount of ceramides found in knockouts may be plotted normalized to the amount of control.

Page 7 line 153: Is mRNA of Cers3 the only mRNA of all expressed CerS1-6 that is decreasing? If possible, please indicate levels of the other Cers expressed in testes. How about Cers5/6 which would influence C16-ceramide levels?

Page 10 line 217 to 220. The authors might compare these findings with similar previous findings.

Page 10 line 235 to page 11 line 240: The intact BTB may be discussed comparing with infertility due to ether lipids, which goes along with a leaky BTB and incorporation of the tracer biotin into the multinucleated giant cells.

Figure 2d: Fluorescent staining with gammaH2AX, SYCP3, and DAPI. Left to the upper pictures, The word "gammaH2AX" should be printed in green and the word "SYCP3" should be printed in red, I guess.

Figure 3a: in PC, the two fatty acids are not directly attached to choline but to glycerophosphatidylcholine. Same holds true for PE and its headgroup.

Figure 4a: metabolic path of VLC-PUFA-Ceramide synthesis. The scheme has to be corrected as in mammals it is not possible to start with saturated fatty acids and convert them with FADS1/2/SCD1 and elongases to PUFA- or VLC-PUFA fatty acids. Minimum starters are the essential fatty acids 18:2 and 18:3. A proper scheme can be found in: Kihara, A. Prog Lipid Res 2016 Vol. 63 Pages 50-69.

Figure 5a and b: 5a does not help to understand b if in b a generalized polarization (GP) index is plotted. This should also be indicated in 5b.

Publication 49 does not allow to find out with what kind of MS/MS mode ceramides have been identified and quantified using C18-UPLC-MS/MS. Please specify if targeted or untargeted. Was it a targeted MRM analysis with transition to m/z264 in order to measure ceramides with a C18-sphingosine as base? Please add this information.

Page 15 line 346: Please add temperature, at which the alkaline hydrolysis was performed.

Page 16 lines 353/354, the indicated supplementary tables for the complete lipid composition data are missing and have to be added.

“Western” in Western blots should be printed with a capital “W”.

Please add a list of abbreviations.

Please change:

“Homolog pairing” should be “homologous pairing”,

“Lipid membrane” (a cellular membrane always contains lipids, but also proteins) should be context dependent “nuclear membrane” or “plasma membrane”,

“lumens” should be “lumina”

Page 6, line 133: please find better word to replace „stark”.

Page 12, line 282 “intercellular bridge” should be in plural

Page 13, line 284, 287 “bridge” should be in plural.

Page 17, line 395 H2Ö should be H2O

Reviewer #2:

Remarks to the Author:

In the manuscript entitled “Regulation of meiotic telomere dynamics through lipid membrane fluidity governed by AdipoR2-ELOVL2”, Zhang and colleagues evaluate how the lipid composition, specifically the contribution of very-long-chain polyunsaturated fatty acid, contributes to meiosis and the formation of germ cell syncytium. The study utilizes a combination of previously established AdipoR2 KO mice, lipidomics, imaging-based evaluation of membrane fluidity, and histological analysis. Through these methods, the authors assert that the lipid makeup of the nuclear envelope (NE) plays a critical role in facilitating appropriate telomeric anchoring, a vital step for ensuring accurate meiotic progression.

Very long-chain polyunsaturated fatty acids have long been known to be crucial for both the maintenance of membrane fluidity and germ cell development, this manuscript attempts to provide evidence for their role in maintaining proper NE structure and organization. Although intriguing the authors fail to provide convincing proof for a direct effect on the NE, that lies at the root of their observation. In addition, a notable issue arises with the figures provided. They appear to depict an earlier or different version from those referenced in the text, creating a disconnect between the indicated figure panels and the actual figures throughout the paper. While this discrepancy does not significantly impede the critical assessment, it does result in a rather exasperating reading experience. Please find a few comments below:

1. AdipoR2 (and AdipoR1) have been previously reported to act as sensors of plasma membrane fluidity. In Figure 5b the Laurdan staining confirms that this observation can be extended to spermatocytes and round spermatids. But, despite the usefulness of Laurdan over other whole cell methods such as lipidomics, this figure fails to provide evidence for AdipoR2 ko mediated “Stiffening” of the Nuclear Envelope. Indeed, the GP index provided is “whole cell.” The authors should consider complementary methods to demonstrate that this is impacting directly on the nucleus, for example AFM on isolated nuclei. The extensive invaginations would suggest otherwise.
2. One could argue “membrane rigidification impairs the nuclear peripheral distribution of meiotic telomeres” is misleading as the authors beautifully demonstrate that the telomeres are correctly attaching to the NE. The problem is not so much an impairment as a gross defect in nuclear structures, resulting in multiple (apparent) Type II invaginations that act as additional anchoring

points for the telomeres. Yet, no effort was made to investigate the alterations in nuclear morphology that lies at the heart of the entire phenotype. Better characterization of this nucleoplasmic-reticulum would greatly strengthen the manuscript. What is the farnesylation state of Lamin B for example?

We thank the Editor and Reviewers for providing constructive comments on our manuscript. Please find below our point-by-point responses to all the reviewer's comments. The text highlighted in red indicates our response.

Reviewer #1:

All somatic cells express in their membranes sphingolipids (ceramides, sphingomyelins and glycosphingolipids), which contain saturated long and saturated or monounsaturated very long acyl chains. In contrast male fertility requires the expression of very unique sphingolipids containing even longer very long and simultaneously also polyunsaturated acyl chains (VLC-PUFA sphingolipids), which are upregulated in spermatocytes during spermatogenesis. Their requirement for spermatogenesis had been proven using knockout mice lacking enzymes responsible for their synthesis (Elovl2, CerS3, Ugcg, B4galnt1). These lipid structures are also found in human testis, indicating corresponding functions. Loss of these sphingolipids leads to an arrest in spermatogenesis, which was shown to go along with formation of multinucleated giant cells and loss of proper intercellular bridges between clonal spermatids. However, detailed mechanisms on the molecular function of these unique sphingolipids still remain elusive as well as regulation of their synthesis.

With this manuscript, the authors add a major puzzle piece to the understanding of the regulation of VLC-PUFA sphingolipids synthesis and give insight into further fundamental cellular processes, impaired by the absence of these unique sphingolipids. Therefore, this manuscript should be of great interest to the readership of Nature Communications as it exemplifies, how specific cellular processes are supported by a correspondingly adjusted lipid membrane composition.

Major comments

To support the conclusions of the authors, it would be very helpful to show by immunofluorescence of wild type testis, which cell types and differentiation stages express AdipoR2 to what kind of degree. Specifically the subcellular localization of AdipoR2 in spermatocytes would be very informative, i.e. is it plasma membrane staining or is it ER/nuclear envelope staining, or both.

We performed immunostaining of AdipoR2 using the rabbit polyclonal antibody, the same one utilized in Fig. 1b, which successfully detected AdipoR2 in Western blotting, demonstrating its high specificity. However, as depicted in the images below (left), there was no specific localization of AdipoR2 in testis seminiferous tubules. Additionally, in an effort to enhance resolution, we attempted immunostaining using isolated spermatocytes; however, once again, there were no specific signals observed (only background staining on XY chromosomes was detected) (right).

To try to address concerns about the antibody possibly having low affinity in immunostaining, we developed new rabbit polyclonal antibodies targeting AdipoR2 (this antibody is not used in the current manuscript). This antibody demonstrated high specificity and high affinity, evident in its efficient immunoprecipitation of AdipoR2 from mouse testis extract (below left). Employing this newly generated antibody, we conducted immunostaining on testis sections. Regrettably, as illustrated in the image below (right), no specific signal was detected again.

Based on these experiments, we now believe that the restricted amount of AdipoR2 protein within the cell poses a challenge for its detection through immunostaining. We note that most publications visualizing AdipoR2 by immunostaining detect highly expressed tagged transgenes, suggesting that the detection of endogenous AdipoR2 is a common problem in the field.

During attachment of telomeres to the nuclear envelope: is there really an internalization of telomeres together with the inner and outer NE components, or is this always an unusual invagination of the nuclear membranes as depicted in figures 5f-h? This should be clarified.

In the TEM analysis, we could not detect any defects in the telomere to the NE attachment as seen by the formation of an intact attachment plate in *AdipoR2* KO spermatocytes (Fig. 5f and 5h). Even in the case of internalized telomeres, we always saw the intact attachment plates formed at the invaginated NE (Fig. 5f and 5h). Thus, we reasonably interpret that the internalized telomeres seen by the cross-section of spermatocytes' immunostaining (Fig. 5e) are all attributed to the NE invagination, in contrast to the situation in *Terb2* KO spermatocytes, where telomeres are internalized without NE invagination due to the defects in the formation of attachment plate (Fig. 5f). I appreciate the reviewer's comment, which made us realize that the previous writing failed to make this point clear. We have changed the conclusive sentence as below.

Before

“Taken together these results suggest that the stiffening of the nuclear membrane in AdipoR2 KO spermatocytes impairs the integrity of the NE leading to defects in the nuclear peripheral distribution of telomeres in meiosis.”

After

“Taken together these results suggest that the stiffening of the lipid membrane in AdipoR2 KO spermatocytes leads to the NE invagination, which disturbs the nuclear peripheral distribution of telomeres in meiosis.”

Furthermore, can such invaginations be explained by stiffening of the lipid membrane? Wouldn't higher curvatures, needed for such invaginations, be in favor of more fluid membranes?

In this context a higher magnification and resolution of TEM pictures should be provided, in addition.

KASH5, a Dynein activation adapter protein, is normally localized to the telomeres (Fig. 5e), suggesting that dynein-dependent forces are equally applied to the telomeres in *AdipoR2* KO spermatocytes. If the NE is fluid enough, these forces are transmitted as telomere movements within the fluid membrane. However, if the NE is less fluid and the telomeres are tightly embedded within the NE, the forces cannot induce movement within the membrane but instead lead to the curvature of the NE, resulting in NE invagination. This is our hypothetical model. We have added this interpretation in the last paragraph of this result section as below.

“It is likely that the dynein-dependent forces applied to telomeres, which normally induce telomere movement within the fluid membrane in WT, instead lead to NE curvature and subsequent invagination in cases where telomeres are tightly embedded within the rigidified membrane.”

Regarding the resolution of the TEM images, we previously provided a compressed figure file to reduce the file size, which is recommended for the 1st submission. We will supply the original high-resolution figures at the time of acceptance.

The interpretation of the authors implies that the VLC-PUFA sphingolipids are constituents of the nuclear membrane. Can the authors add any data supporting this assumption?

Technically, investigating the subcellular localization of specific lipid species is quite challenging, and we currently lack direct evidence indicating that VLC-PUFA sphingolipids are constituents of the NE. This will pose a challenge for future research. As a supporting piece of evidence, a recently published paper also discussed the necessity of unsaturated fatty acids for the architecture and function of the NE in yeast cells (PMID: 37591950). We have cited this paper in the discussion section as follows.

“The necessity of unsaturated FAs for NE architecture appears to be evolutionarily conserved, as observed in recent findings highlighting their importance for the NE structure and function in yeast cells³⁶.”

³⁶ Romanauska, A. & Kohler, A. Lipid saturation controls nuclear envelope function. *Nat Cell Biol* 25, 1290-1302 (2023). <https://doi.org/10.1038/s41556-023-01207-8>

Finding spermatocytes more luminal than round spermatids indicates defects in normal tubular structuring. However, does that really proof an impaired synchronicity within one clone of maturing germ cells? For this, I guess, it would be necessary to demonstrate that these spermatocytes and round spermatids (Fig. 6e) are within one syncytium, i.e. not separated by cell membranes, which would require analysis with electron microscopy. It may very well be, that after formation of multinucleated giant cells, the spermatids included continue to differentiate in a synchronous fashion until a certain point, i.e. initiation of acrosome formation. This is a very reasonable suggestion, which improves the accuracy of our paper. We appreciate your point. Indeed, the defects shown in Fig. 6d and 6e cannot always attributed to the defects in synchronicity in germ cell development, rather can be attributed to the defects in tubular structuring caused by the defective intercellular bridge formation. To prove the “defects in synchronicity”, we should be able to provide evidence showing a single syncytium containing both spermatids and spermatocytes via TEM analysis as suggested by the reviewer. So far, our TME analysis only observed abnormal syncytium containing multiple spermatids as shown in Fig. 1f but not such a mixture of different cell types. Therefore, we have toned down our statement in summary and results as exemplified below.

Before

“Further, the stiffened membrane impairs the formation of intercellular bridges and the germ cell syncytium, which disrupts the synchronous development of male germ cells.”

After revision

“Further, the stiffened membrane impairs the formation of intercellular bridges and the germ cell syncytium, which disrupts the orderly arrangement of cell types within the seminiferous tubules.”

Before

“Thus, we conclude that the synchronicity of germ cell development was impaired following the destabilization of intercellular bridges in AdipoR2 KO seminiferous tubules.”

After revision

“Thus, we conclude that the orderly arrangement of cell types within the seminiferous tubules was impaired following the destabilization of intercellular bridges in AdipoR2 KO seminiferous tubules.”

Minor comments

The authors mixed up all links for figures in the main text with the numbering of figures and extended figures. Some extended figures appear as main figures. This made reading quite difficult and has to be corrected. I do not understand how 13 authors could have missed to see this. Extended Figure 2e, as mentioned in the main text, does not exist at all.

I apologize for any inconvenience my previous numbering errors may have caused the reviewers. I have now corrected all the figure numbering in the revised version.

Page 4 lines 88/89: The statement is not proven by Fig. 1d as the resolution of Fig. 1d does not allow the reader to judge if MGs are filled with nuclei of spermatocytes or spermatids and not of which stage spermatocytes could be.

*We now provide high-resolution and bigger images in Supplementary Fig. 2, where we can clearly observe that the multinucleated giant round spermatids (RSs) are filled with nuclei much smaller than those of spermatocytes. Notably, these nuclei are even smaller than the nuclei of surrounding solitary RSs. These abnormal multinucleated giant RSs are undergoing apoptosis, as shown in Fig. 1e, which is likely the reason for the abnormal nuclear shape and size in these multinucleated giant RSs. Beyond the hematoxylin and eosin staining pictures, **the reason we can confidently conclude that these multinucleated giant cells are RSs is that we could distinctly observe the basal body structures and the thickened nuclear envelope at the apical side in the transmission electron microscopy analysis** (as described in Fig. 1f and the corresponding figure legend), both of which are hallmarks of RSs and are never seen in spermatocytes.*

Supplementary Fig.2 : Spermatogenesis defects in *AdipoR2*^{-/-} seminiferous tubules

Testis sections from 8-week-old WT and *AdipoR2*^{-/-} males stained with hematoxylin and eosin. The arrowheads indicate a spermatocyte (SPC), round spermatid (RS), elongated spermatid (ES), and multinucleated giant round spermatid (MGRS). Scale bars: 50 μ m.

Page 6 lines 136/137: I consider a complete loss of a lipid more significant than an increase of a lipid by 22 mol%. Moreover, the loss of ceramides with VLC-PUFAs automatically has to be compensated in an increase of another ceramide species as the representation of mol% will always add up to 100%. Hence: Presentation of lipid data in mol% may be misleading and does not allow to conclude if a subspecies is down or up regulated. Therefore absolute or exact lipid concentrations have to be plotted, which allow the reader to judge if, i.e. a ceramide with palmitic acid is really upregulated or its mol% simply increases, because ceramides with PUFAs are downregulated. If this is not possible, the relative amount of ceramides found in knockouts may be plotted normalized to the amount of control.

Thank you for bringing this to our attention. We believed that measuring tissue weight, which is necessary for normalizing the absolute concentration of each lipid species, might be less accurate compared to the highly sensitive lipidomics analysis. Therefore, we chose to present the graph in mol% in the main figure. In response to the reviewer's request, we have included a Supplementary Fig. 4 showing the absolute amount of each lipid species (in nmol or pmol), normalized to the tissue weight (in mg). The data exhibits the same trends as the graph displaying mol%. We now mention this data in the result section.

Page 7 line 153: Is mRNA of *Cers3* the only mRNA of all expressed *CerS1-6* that is decreasing? If possible, please indicate levels of the other *Cers* expressed in testes. How about *Cers5/6* which would influence C16-ceramide levels?

In a previous study (PMID: 18308723, Fig 4H), the expression levels of *CerS1-6* were examined during murine testicular development. Only *CerS3* expression correlated with the increasing abundance of VLC-PUFA glycosphingolipids during development, suggesting that the other *CerS* enzymes are not involved in the testis-specific VLC-PUFA synthesis. Indeed, male infertility was not reported in *CerS1* (PMID: 23074226), *CerS2* (PMID: 19801672), *CerS4* (PMID: 24738593), *CerS5* (PMID: 26853464), and *CerS6* (PMID: 23760501) KO mice, and only *CerS3* conditional KO mice (PMID: 26045466) showed defects in male reproduction.

To further experimentally support our hypothesis that the reduction of other *CerS* enzymes is not the cause of lipidome defects observed in *AdipoR2* KO mice, we conducted qPCR analysis for *CerS1*, *CerS2*, *CerS4*, *CerS5*, and *CerS6*, as presented in Supplementary Fig. 6a. The results indicated that there was no significant decrease in the expression of these enzymes. These findings strengthen our conclusion that the

expression of *CerS3* is specifically impaired in *AdipoR2* KO mice. We now mention this data in the result section.

Page 10 line 217 to 220. The authors might compare these findings with similar previous findings.

Indeed, similar defects in TEX14 ring structures were reported in germ cell-specific conditional knockouts of *CerS3* and *Gcs*. We have added the following sentence in the results section with the citation of the following paper.

“Similar defects in TEX14 integrity were reported in germ cell-specific conditional knockouts of *CerS3* and *Gcs* (a gene encoding a glucosylceramide synthase responsible for the addition of glycan moieties to sphingolipids)⁸, suggesting that the defects in the intercellular bridge integrity are hallmark issues caused by abnormal sphingolipid and glycosphingolipid composition in male germ cells.”

⁸ Rabionet, M. et al. Male meiotic cytokinesis requires ceramide synthase 3-dependent sphingolipids with unique membrane anchors. *Hum Mol Genet* 24, 4792-4808 (2015). <https://doi.org/10.1093/hmg/ddv204>

Page 10 line 235 to page 11 line 240: The intact BTB may be discussed comparing with infertility due to ether lipids, which goes along with a leaky BTB and incorporation of the tracer biotin into the multinucleated giant cells.

Thank you for pointing this out. We have added this discussion in the result section as shown below by citing the corresponding preceding study.

“The proper composition of lipid species in the testis is essential for the integrity of the BTB, as demonstrated by its disruption in ether lipid-deficient mice²⁸.”

²⁸ Komljenovic, D. et al. Disruption of blood-testis barrier dynamics in ether-lipid-deficient mice. *Cell Tissue Res* 337, 281-299 (2009). <https://doi.org/10.1007/s00441-009-0809-7>

Figure 2d: Fluorescent staining with gammaH2AX, SYCP3, and DAPI. Left to the upper pictures, The word “gammaH2AX” should be printed in green and the word “SYCP3” should be printed in red, I guess. Thank you for bringing this to our attention. We have corrected the mistake.

Figure 3a: in PC, the two fatty acids are not directly attached to choline but to glycerophosphatidylcholine. Same holds true for PE and its headgroup.

Thank you for bringing this to our attention. We have made the following changes to the illustration to improve its accuracy.

Figure 4a: metabolic path of VLC-PUFA-Ceramide synthesis. The scheme has to be corrected as in mammals it is not possible to start with saturated fatty acids and convert them with FADS1/2/SCD1 and elongases to PUFA- or VLC-PUFA fatty acids. Minimum starters are the essential fatty acids 18:2 and 18:3. A proper scheme can be found in: Kihara, A. Prog Lipid Res 2016 Vol. 63 Pages 50-69. **We have made the following changes to the illustration to improve its accuracy.**

Figure 5a and b: 5a does not help to understand b if in b a generalized polarization (GP) index is plotted. This should also be indicated in 5b.

Thank you for pointing this out. We have colored the Fig. 5a schematics to match the color with the pseudocolor shown in Fig. 5b.

Publication 49 does not allow to find out with what kind of MS/MS mode ceramides have been identified and quantified using C18-UPLC-MS/MS. Please specify if targeted or untargeted. Was it a targeted MRM analysis with transition to m/z 264 in order to measure ceramides with a C18-sphingosine as base? Please add this information

We have added the below sentence describing the detailed method.

“In brief, the separation of ceramides species were performed using a Waters BEH C8 column (2.1×100 mm with $1.7\mu\text{m}$ particles) with water, acetonitrile and isopropanol as mobile phases. The detection was made in positive MRM (targeted) mode using m/z 264 as fragment ion.”

Page 15 line 346: Please add temperature, at which the alkaline hydrolysis was performed.

We have added the information.

“prior to analysis the lipid extracts were evaporated and exposed to alkaline hydrolysis (0.1 M KOH in methanol for 60 minutes **at room temperature**) in order to remove the interfering phospholipids.”

Page 16 lines 353/354, the indicated supplementary tables for the complete lipid composition data are missing and have to be added.

We have now provided the requested data, along with all the source data, as a source data file “Source data.xlsx”.

“Western” in Western blots should be printed with a capital “W”.

Thank you for bringing this to our attention. We have corrected the mistake.

Please add a list of abbreviations.

We have added the list of abbreviations following the abstract.

Please change:

“Homolog pairing” should be “homologous pairing”,

Fixed

“Lipid membrane” (a cellular membrane always contains lipids, but also proteins) should be context dependent “nuclear membrane” or “plasma membrane”,

Fixed

“lumens” should be “lumina”

Fixed

Page 6, line 133: please find better word to replace „stark“.

Fixed (we removed the whole sentence)

Page 12, line 282 “intercellular bridge” should be in plural

Fixed

Page 13, line 284, 287 “bridge” should be in plural.

Fixed

Page 17, line 395 H₂O should be H₂O

Fixed

Reviewer #2:

In the manuscript entitled “Regulation of meiotic telomere dynamics through lipid membrane fluidity governed by AdipoR2-ELOVL2”, Zhang and colleagues evaluate how the lipid composition, specifically the contribution of very-long-chain polyunsaturated fatty acid, contributes to meiosis and the formation of germ cell syncytium. The study utilizes a combination of previously established AdipoR2 KO mice, lipidomics, imaging-based evaluation of membrane fluidity, and histological analysis. Through these methods, the authors assert that the lipid makeup of the nuclear envelope (NE) plays a critical role in facilitating appropriate telomeric anchoring, a vital step for ensuring accurate meiotic progression.

Very long-chain polyunsaturated fatty acids have long been known to be crucial for both the maintenance of membrane fluidity and germ cell development, this manuscript attempts to provide evidence for their role in maintaining proper NE structure and organization. Although intriguing the authors fail to provide convincing proof for a direct effect on the NE, that lies at the root of their observation. In addition, a notable issue arises with the figures provided. They appear to depict an earlier or different version from those referenced in the text, creating a disconnect between the indicated figure panels and the actual figures throughout the paper. While this discrepancy does not significantly impede the critical assessment, it does result in a rather exasperating reading experience. Please find a few comments below:

Thank you so much for your effort in evaluating our manuscript. We sincerely apologize for the trouble caused by the disconnection between the figure numbers and the corresponding text. It appears that errors occurred during the transfer process between the two journals, resulting in the submission of an older version of the manuscript. We greatly appreciate your time and effort in reading and evaluating our manuscript.

1. AdipoR2 (and AdipoR1) have been previously reported to act as sensors of plasma membrane fluidity. In Figure 5b the Laurdan staining confirms that this observation can be extended to spermatocytes and round spermatids. But, despite the usefulness of Laurdan over other whole cell methods such as lipidomics, this figure fails to provide evidence for AdipoR2 ko mediated “Stiffening” of the Nuclear Envelope. Indeed, the GP index provided is “whole cell.” The authors should consider complementary methods to demonstrate that this is impacting directly on the nucleus, for example AFM on isolated nuclei. The extensive invaginations would suggest otherwise.

Technically, investigating the lipid composition or fluidity of specific membrane types, such as the nuclear envelope within the cell, is quite challenging. Currently, we lack direct evidence indicating that VLC-PUFA sphingolipids are constituents of the NE, and the specific impact of *AdipoR2* KO on the fluidity of the NE in the testis has not been directly tested. This presents a challenge for future research. While

acknowledging the potential merit of exploring AFM on isolated nuclei for a more direct assessment of nuclear envelope, regrettably, resource limitations, including time and equipment availability, preclude the feasibility of conducting this additional experiment in the present study.

As a supporting piece of evidence, a recently published paper also discussed the necessity of unsaturated fatty acids for the architecture and function of the NE in yeast cells (PMID: 37591950). We have cited this paper in the discussion section as follows.

“The necessity of unsaturated FAs for NE architecture appears to be evolutionarily conserved, as observed in recent findings highlighting their importance for the NE structure and function in yeast cells³⁶.”

³⁶Romanauska, A. & Kohler, A. Lipid saturation controls nuclear envelope function. Nat Cell Biol 25, 1290-1302 (2023). <https://doi.org/10.1038/s41556-023-01207-8>

Regarding the concern about Laurdan dye staining, indeed, Laurdan dye typically stains all cellular membranes, as shown in the pictures below. Thus, we quantified the signals overlapping with the entire cellular region, encompassing signals derived from all types of cellular membranes, including the plasma membrane and the NE.

However, in very rare instances, we could find spermatocytes with clearly visible Laurdan dye signals originating from the NE, as shown in the pictures below. In these cells, the pseudocolor at the NE exhibited the same tendency as the plasma membrane or the entire cellular signals, suggesting that not only the plasma membrane but also the NE is rigidified in *AdipoR2* KO spermatocytes. However, given the rarity of cells displaying the NE signal (less than 10% of the observed cells) and the difficulty in quantifying the thin NE signals without being affected by the surrounding background signals, it was considered challenging to conduct an objective quantitative analysis specifically focusing on the NE signals. Therefore, for the sake of accuracy at the present stage, these data are not included in the paper.

2. One could argue “membrane rigidification impairs the nuclear peripheral distribution of meiotic telomeres” is misleading as the authors beautifully demonstrate that the telomeres are correctly attaching to

the NE. The problem is not so much an impairment as a gross defect in nuclear structures, resulting in multiple (apparent) Type II invaginations that act as additional anchoring points for the telomeres. Yet, no effort was made to investigate the alterations in nuclear morphology that lies at the heart of the entire phenotype. Better characterization of this nucleoplasmic-reticulum would greatly strengthen the manuscript. What is the farnesylation state of Lamin B for example?

Thank you for your insightful comments and valuable suggestions. We acknowledge your discussion. Indeed, as pointed out, the attachment of telomeres to the NE remains intact in the absence of AdipoR2, as seen by the intact attachment plates. This holds true even for the telomeres attached to the invaginated NE, as shown in Fig. 5h. This observation is reasonable since the proteins associated with telomere-NE attachment, such as TERB1, TERB2, MAJINA, SUN1, and KASH5, were localized to the meiotic telomeres in a manner similar to the WT situation.

Our data overall support the idea that the invagination of NE and the subsequent internalization of telomeres were caused by the change in the fatty acid composition in the membrane lipid, particularly the loss of VLC-PUFA, as indicated by the lipidomics analysis in Fig. 3. Furthermore, we demonstrated that the loss of VLC-PUFA can be reasonably explained by the reduction in ELOVL2 protein expression in the mutant testes (Fig. 4). Our findings are further supported by the fact that some of the defects in *AdipoR2* KO testes resemble those reported in *Elovl2* KO testes (PMID: 21106902).

Overall, according to our analysis, the primary defects are related to lipid composition rather than nuclear morphology and Lamin B farnesylation. We cannot rule out the possibility that Lamin B farnesylation could be affected as a secondary defect; however, we believe it is beyond the scope of the current manuscript, which mainly focuses on the membrane lipid aspect.

To directly connect the lipidomics defects with the observed NE invagination, we now introduce a discussion model. If the NE is less fluid and the telomeres are tightly embedded within it, the forces generated may not induce telomere movements within the membrane. Instead, they could lead to the curvature of the NE, resulting in NE invagination. This hypothetical model has now been incorporated into the results section as follows.

“It is likely that the dynein-dependent forces applied to telomeres, which normally induce telomere movement within the fluid membrane in WT, instead lead to NE curvature and subsequent invagination in cases where telomeres are tightly embedded within the rigidified membrane.”

Reviewers' Comments:

Reviewer #1:

Remarks to the Author:

Minor comments:

Abstract line 31

Instead of writing "...stiffens the cellular membrane, causing the invagination..." I would suggest "...stiffens the cellular membrane and causes the..". Reason: It is clear that loss of VLC-PUFA is the cause for both, however it is not proven that the stiffened membrane causes invaginations, although it may likely be.

Abstract line 35

Instead of writing "Our findings propose..." I would prefer "According to our findings we propose..."

Reviewer #2:

Remarks to the Author:

The authors have addressed the criticisms.

We thank the Reviewers for providing constructive comments on our manuscript.

We have incorporated the additional suggestions provided by the reviewers and editors. The abstract has been modified in line with the reviewer's recommendations. Furthermore, we have expanded the introduction to provide a more comprehensive background and adjusted the figure along with its corresponding legend to comply with the editorial requests.